# GRAPH-R1: TOWARDS AGENTIC GRAPHRAG FRAMEWORK VIA END-TO-END REINFORCEMENT LEARNING

## ABSTRACT

Retrieval-Augmented Generation (RAG) mitigates hallucination in LLMs by incorporating external knowledge, but relies on chunk-based retrieval that lacks structural semantics. GraphRAG methods improve RAG by modeling knowledge as entity-relation graphs, but still face challenges in high construction cost, fixed one-time retrieval, and reliance on long-context reasoning and prompt design. To address these challenges, we propose `Graph-R1`, an agentic GraphRAG framework via end-to-end reinforcement learning (RL). It introduces lightweight knowledge hypergraph construction, models retrieval as a multi-turn agent-environment interaction, and optimizes the agent process via an end-to-end reward mechanism. Experiments on standard RAG datasets show that Graph-R1 outperforms traditional GraphRAG and RL-enhanced RAG methods in reasoning accuracy, retrieval efficiency, and generation quality. Our code is publicly available[1].

## 1 INTRODUCTION

Large Language Models (LLMs) (Zhao et al., 2025a) have achieved widespread success in NLP tasks. However, when applied to knowledge-intensive or proprietary knowledge-dependent applications, they still suffer from the hallucination problem (Zhang et al., 2023), generating inaccurate content. To improve credibility and factual consistency, Retrieval-Augmented Generation (RAG) (Lewis et al., 2020) introduces external knowledge sources as references, alleviating the knowledge bottleneck of pure language modeling. Nevertheless, existing RAG methods mostly rely on chunk-based text

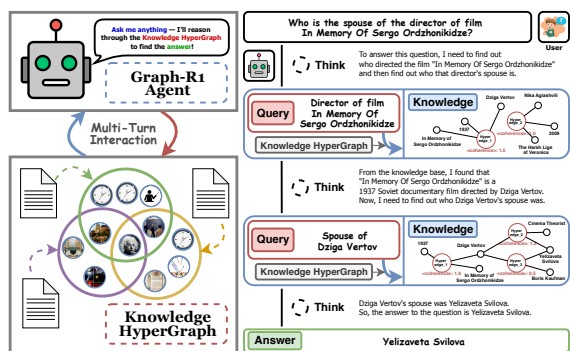

Figure 1: An illustration of Graph-R1.

blocks (Gao et al., 2024), which makes it difficult to capture complex knowledge structures among entities. To address this, GraphRAG methods (Edge et al., 2025; Guo et al., 2025; Luo et al., 2025a) represent knowledge as entity-relation graphs, enhancing retrieval efficiency and generation quality.

Generally, GraphRAG methods consist of three processes: *knowledge graph construction*, *graph retrieval*, and *answer generation*. First, knowledge graphs are typically constructed by LLMs to extract entities and relations from text, forming a graph structure (Xu et al., 2024). Second, the retrieval process queries relevant subgraphs or paths through subgraph retrieval or path pruning strategies (Chen et al., 2025; Gutiérrez et al., 2025). Finally, the generation process prompts LLMs to generate answers based on the retrieved graph-based knowledge (Xiao et al., 2025).

However, current GraphRAG methods still face three key challenges: **(i) High cost and semantic loss in *knowledge construction* process.** Compared to standard RAG, GraphRAG methods convert natural language knowledge into graph structures using LLMs, which results in high cost and often causes semantic loss relative to the original content (Luo et al., 2024; 2025a). **(ii) Fixed retrieval process with only one-time interaction in *graph retrieval* process.** Although existing GraphRAG

---

[1]Anonymous Github Link: `https://anonymous.4open.science/r/Graph-R1`

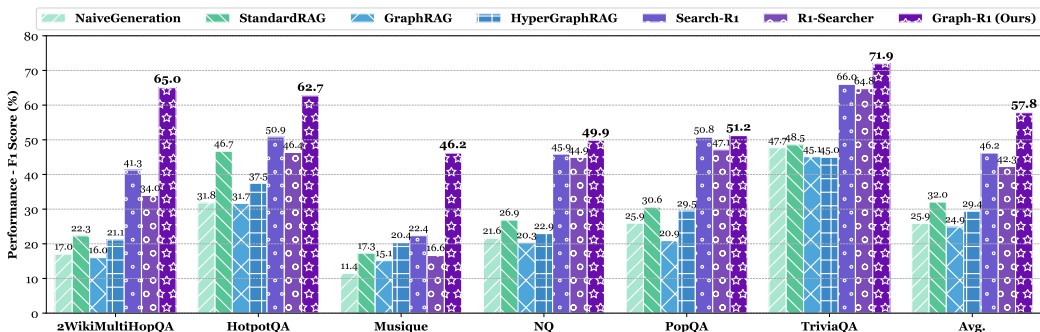

Figure 2: Comparison of F1 scores across RAG benchmarks. Using a graph as the knowledge environment enables RL to achieve a higher performance ceiling compared to chunk-based knowledge.

methods design various retrieval strategies to improve efficiency, most of them aim to gather sufficient knowledge in a single fixed retrieval (Chen et al., 2025), which limits performance in complex queries. **(iii) Dependence on large LLMs for long-context analysis and prompt quality in *answer generation* process.** Generation based on retrieved graph-structured knowledge often requires strong long-context reasoning ability, making the output quality highly dependent on the LLM's parameter size and prompt design (Guo et al., 2025), resulting in unstable reasoning and generation.

To address these challenges, we propose `Graph-R1`, as illustrated in Figure 1, an agentic GraphRAG framework enhanced by end-to-end reinforcement learning (RL), inspired by DeepSeek-R1 (DeepSeek-AI et al., 2025). First, we propose a lightweight knowledge hypergraph construction method to establish a standard agent environment for the query action space. Moreover, we model the retrieval process as a multi-turn agentic interaction process, enabling LLMs to repeatedly perform the reasoning loop of "think-retrieve-rethink-generate" within the knowledge hypergraph environment. Furthermore, we design an end-to-end reward mechanism that integrates generation quality, retrieval relevance, and structural reliability of graph paths into a unified optimization objective. Using RL, the agent learns a generalizable graph reasoning strategy and achieves tighter alignment between structured graph-based knowledge and language generation.

We perform experiments on various standard RAG datasets (Jin et al., 2025b). Experimental results demonstrate that Graph-R1 outperforms traditional GraphRAG methods and RAG combined with RL methods (Jin et al., 2025a; Song et al., 2025) in reasoning accuracy, retrieval efficiency, and generation quality. As shown in Figure 2, the end-to-end RL strategy guides the agent through multiple turns of interaction and goal-driven exploration in the graph, effectively bridging the gap between knowledge representation and language generation. This work lays a foundation for building the next generation of knowledge-driven and strategy-optimized agent-based generation systems.

## 2 RELATED WORK

**RAG and GraphRAG.** Retrieval-Augmented Generation (RAG) (Lewis et al., 2020) improves LLM factuality by retrieving external knowledge, but struggles with data silos and limited structural reasoning. GraphRAG (Edge et al., 2025) mitigates these issues by incorporating graph-structured knowledge, inspiring enterprise systems (Wu et al., 2024; Liang et al., 2025; Wang et al., 2025a) and efficient variants like LightRAG (Guo et al., 2025). Recent works further enhance representation power using hypergraphs, causal graphs, heterogeneous graphs, or hierarchical graphs (Luo et al., 2025a; Feng et al., 2025b; Wang et al., 2025b; Xu et al., 2025; Zhuang et al., 2025), while retrieval optimization explores path-based exploration and pruning techniques (Chen et al., 2025; Gutiérrez et al., 2025; Liu et al., 2025; Wang, 2025; Zhao et al., 2025b; Luo et al., 2025c;d). Although most existing GraphRAG systems still follow a single-shot retrieval paradigm, recent methods (Lee et al., 2025; Dong et al., 2025; Yang et al., 2025b) attempt multi-step retrieval via LLM-driven query refinement. However, these strategies remain prompt-dependent, lacking a learnable policy that interacts with graph structures. In this work, we propose Graph-R1, the first agentic GraphRAG framework that learns a multi-turn retrieval policy end-to-end. Concurrently, Yu et al. (2025) applies a similar optimization within GraphRAG but operates without using knowledge hypergraphs.

**Reinforcement Learning for LLMs.** Reinforcement learning (RL) is increasingly adopted to enhance LLM reasoning (Wu, 2025; Luo et al., 2025b), as demonstrated by OpenAI's o1/o3/o4 (OpenAI et al., 2024b). DeepSeek-R1 (DeepSeek-AI et al., 2025) achieves comparable capabilities and further introduces the Group Relative Policy Optimization (GRPO) (Shao et al., 2024) for scalable end-to-end training. GRPO-based reasoning has been extended to tasks such as visual understanding(Shen et al., 2025), logical reasoning (Xie et al., 2025), and program synthesis (Ma et al., 2025). RL-enhanced agents have also shown strong performance in multi-turn interaction (Lu et al., 2025; Feng et al., 2025a) and open-domain retrieval (Jin et al., 2025a; Song et al., 2025; Zheng et al., 2025; Sun et al., 2025), highlighting RL's potential in agentic GraphRAG frameworks (Gao et al., 2025).

## 3 Preliminaries

We formalize the GraphRAG pipeline into three stages as detailed below:

**(a) Knowledge Graph Construction.** This stage extracts structured relational facts from raw text. Given a knowledge collection $K = \{d_1, d_2, \ldots, d_N\}$, the goal is to extract facts $f_d$ from each semantic unit $d \in K$ and aggregate them into a unified graph $\mathcal{G}_K$:

$$\mathcal{G}_K \sim \sum_{d \in K} \pi_{\text{ext}}(f_d \mid d), \tag{1}$$

where $\pi_{\text{ext}}$ denotes an LLM-based extractor that parses each $d$ into a set of relation-entity pairs $f_d = \{(r_i, \mathcal{V}_{r_i})\}$, with $r_i$ as the relation and $\mathcal{V}_{r_i} = \{v_1, \ldots, v_n\}$ the participating entities.

**(b) Graph Retrieval.** Graph retrieval is formulated as a two-step process over $\mathcal{G}_K$: (1) retrieving candidate reasoning paths and (2) pruning irrelevant ones. Conditioned on a query $q$, the model first retrieves a candidate set $\mathcal{X}_q = \{x_1, \ldots, x_m\}$ and then selects a relevant subset $\mathcal{Z}_q \subseteq \mathcal{X}_q$. The overall objective is to maximize the expected joint likelihood of the two steps:

$$\max_{\theta} \ \mathbb{E}_{\mathcal{Z}_q \sim P(\mathcal{Z}_q \mid q, \mathcal{G}_K)} \left[ \prod_{t=1}^{T_x} P_\theta(x_t \mid x_{<t}, q, \mathcal{G}_K) \cdot \prod_{t=1}^{T_z} P_\theta(z_t \mid z_{<t}, \mathcal{X}_q, q) \right], \tag{2}$$

where $T_x$ and $T_z$ denote the number of retrieved and selected paths, respectively.

**(c) Answer Generation.** Given a query $q$ and selected paths $\mathcal{Z}_q$, answer generation produces a natural language answer $y$ grounded in graph-based evidence, formulated as:

$$P(y \mid q, \mathcal{G}_K) = \sum_{\mathcal{Z}_q \subseteq \mathcal{X}_q} P(y \mid q, \mathcal{Z}_q) \cdot P(\mathcal{Z}_q \mid q, \mathcal{G}_K), \tag{3}$$

where $P(y \mid q, \mathcal{Z}_q)$ is generation likelihood and $P(\mathcal{Z}_q \mid q, \mathcal{G}_K)$ is retrieval-pruning distribution.

## 4 Methodology: Graph-R1

In this section, as illustrated in Figure 3, we introduce Graph-R1, including agent initialization, multi-turn graph interaction, and outcome-directed end-to-end reinforcement learning.

### 4.1 Knowledge Construction and Agent Initialization

Graph-R1 adopts an LLM-driven agent, initialized with a knowledge hypergraph environment $\mathcal{G}_H$, the action space $\mathcal{A}$, the state space $\mathcal{S}$, and the answer target $y_q$ for the given query $q$.

**Graph Environment $\mathcal{G}_H$.** To support agentic reasoning, we propose a lightweight method for constructing a knowledge hypergraph $\mathcal{G}_H$ from given domain knowledge $K = \{d_1, d_2, \ldots, d_N\}$. For each chunk unit $d \in K$, an LLM-based extractor $\pi_{\text{ext}}$ identifies $m$ n-ary relational facts, where each comprises a semantic segment $h_i$ and a set of participating entities $\mathcal{V}_{h_i} = \{v_1, \ldots, v_n\}$. A shared encoder $\phi(\cdot)$ is then used to generate semantic embeddings for both entities and relations:

$$\mathcal{G}_H = (V, E_H, \phi), \quad \text{where} \quad \pi_{\text{ext}}(d) \to \{(h_i, \mathcal{V}_{h_i})\}_{i=1}^m, \ \phi(v) = \text{Enc}(v), \ \phi(h_i) = \text{Enc}(h_i), \tag{4}$$

where each $h_i$ defines a hyperedge $h_i \in E_H$ connecting its associated entities $\mathcal{V}_{h_i}$ as $v \in V$. The resulting hypergraph $\mathcal{G}_H$ encodes high-order relational structures with rich semantic grounding.

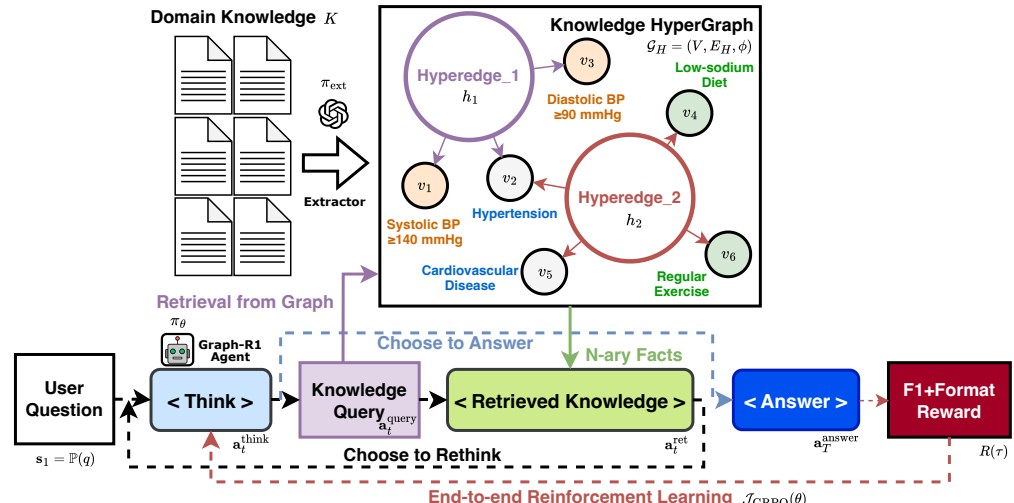

Figure 3: Overview of the Graph-R1 framework: an RL-enhanced reasoning trajectory over knowledge hypergraph, where the agent iteratively decides to think, query, retrieve knowledge, and answer.

**The Agent Action Space** $\mathcal{A}$. In Graph-R1, each agent action $\mathbf{a}_t \in \mathcal{A}$ comprises four sub-actions: *Thinking* $\mathbf{a}_t^{\text{think}}$, which decides whether to continue or terminate reasoning; *Query Generation* $\mathbf{a}_t^{\text{query}}$, which formulates a retrieval query; *Graph Retrieval* $\mathbf{a}_t^{\text{ret}}$, which extracts relevant knowledge from the hypergraph; and *Answering* $\mathbf{a}_t^{\text{ans}}$, which produces a final response if reasoning ends. The agent action $\mathbf{a}_t$ has two compositional forms, and the joint action log-likelihood is defined as:

$$\log \pi(\mathbf{a}_t \mid \mathbf{s}_t) = \begin{cases} \log \mathcal{G}_H(\mathbf{a}_t^{\text{ret}} \mid \mathbf{s}_t, \mathbf{a}_t^{\text{think}}, \mathbf{a}_t^{\text{query}}) + \log \pi(\mathbf{a}_t^{\text{query}} \mid \mathbf{s}_t, \mathbf{a}_t^{\text{think}}) + \\ \log \pi(\mathbf{a}_t^{\text{think}} \mid \mathbf{s}_t), & \text{if } \mathbf{a}_t^{\text{think}} \to \text{continue,} \\ \log \pi(\mathbf{a}_t^{\text{ans}} \mid \mathbf{s}_t, \mathbf{a}_t^{\text{think}}) + \log \pi(\mathbf{a}_t^{\text{think}} \mid \mathbf{s}_t), & \text{if } \mathbf{a}_t^{\text{think}} \to \text{terminate,} \end{cases} \quad (5)$$

where, at each step, the agent first performs *Thinking*, and then conditionally chooses between continuing reasoning (*Query Generation* and *Graph Retrieval*) or terminating via *Answering*.

**The Agent State Space** $\mathcal{S}$ **and Target** $y_q$. At each step $t$, the state $\mathbf{s}_t \in \mathcal{S}$ is defined as $\mathbf{s}_t = (\mathbf{s}_1, \mathbf{a}_1, \ldots, \mathbf{a}_{t-1})$, with $\mathbf{s}_1$ initialized from the input query $q$. Once a termination action $\mathbf{a}_T$ is issued, the agent reaches final state $\mathbf{s}_T$, where $T$ is the total number of reasoning steps, and an answer $y_q \sim \mathbf{a}_T^{\text{ans}}$ is produced to address $q$.

**Proposition 1.** *Graph-structured knowledge boosts agent accuracy by richer representation.*

*Proof.* We provide experimental results in Section 5.2 and theoretical proofs in Appendix B.1. □

### 4.2 KNOWLEDGE REASONING VIA MULTI-TURN GRAPH INTERACTION

We model reasoning as a multi-turn interaction between an agent $\pi_\theta$ and a hypergraph $\mathcal{G}_H$. We first define the step-wise policy $\pi_\theta(\cdot \mid \mathbf{s}_t)$ prompted by Table 1, then describe how to retrieve knowledge $\mathcal{G}_H(\mathbf{a}_t^{\text{ret}} \mid \cdot, \mathbf{a}_t^{\text{query}})$ based on $\mathbf{a}_t^{\text{query}}$ in each step, and finally present the objective to optimize $P(y_q \mid \cdot)$.

**Modeling the Step-wise Reasoning Policy.** At each reasoning step $t$, the LLM governs the agent's behavior by generating a structured output consisting of: (i) a thinking reflection $\mathbf{a}_t^{\text{think}}$ that summarizes the current state and highlights potential knowledge gaps; (ii) a composition indicator $\alpha_t \in \mathcal{A}_{\text{type}} = \{(\texttt{query}, \texttt{retrieve}), (\texttt{answer})\}$ that determines the sub-action structure; and (iii) a content output $\mathbf{a}_t^{\text{out}} \in \mathcal{A}_{\text{content}}$, representing either a retrieval query or a final answer. We model this decision-making process as a hierarchical policy conditioned on the agent state $\mathbf{s}_t \in \mathcal{S}$, which encodes the history of prior actions and retrieved information. The policy is factorized as:

$$\pi_\theta(\mathbf{a}_t^{\text{think}}, \alpha_t, \mathbf{a}_t^{\text{out}} \mid \mathbf{s}_t) = \pi_\theta(\mathbf{a}_t^{\text{out}} \mid \alpha_t, \mathbf{a}_t^{\text{think}}, \mathbf{s}_t) \cdot \pi_\theta(\alpha_t \mid \mathbf{a}_t^{\text{think}}, \mathbf{s}_t) \cdot \pi_\theta(\mathbf{a}_t^{\text{think}} \mid \mathbf{s}_t), \quad (6)$$

where $\pi_\theta$ denotes the LLM-parameterized policy, which encourages three aligned behaviors: generating reflections $\mathbf{a}_t^{\text{think}}$ that assess knowledge sufficiency, selecting $\alpha_t$ to balance exploration and termination, and producing $\mathbf{a}_t^{\text{out}}$ that advances retrieval $\mathbf{a}_t^{\text{query}}$ or yields a direct answer $\mathbf{a}_t^{\text{ans}}$.

---

You are a helpful assistant. Answer the given question. You can query from knowledge base provided to you to answer the question. You can query knowledge as many times as you want. You must first conduct reasoning inside **`<think>`**...**`</think>`**. If you need to query knowledge, you can set a query statement between **`<query>`**...**`</query>`** to query from knowledge base after **`<think>`**...**`</think>`**. When you have the final answer, you can output the answer inside **`<answer>`**...**`</answer>`**. Question: question. Assistant:

---

Table 1: Template for Graph-R1. question will be replaced with the specific user query. Note that the knowledge retrieved is placed within **`<knowledge>`**...**`</knowledge>`** after **`</query>`**.

**Knowledge Interaction via Hypergraph Retrieval.** Given a query $\mathbf{a}_t^{\text{query}}$ generated by the reasoning LLM, we retrieve relevant knowledge $\mathbf{a}_t^{\text{ret}}$ from the hypergraph $\mathcal{G}_H = (V, E_H)$ through a dual-path interaction process: entity-based retrieval and direct hyperedge retrieval. The resulting n-ary relational facts are then aggregated via rank-based fusion to support downstream reasoning.

*(i) Entity-based Hyperedge Retrieval.* We first identify a set of top-ranked entities based on their similarity to the extracted entities $V_{\mathbf{a}_t^{\text{query}}}$, and collect hyperedges that connect to any retrieved entity:

$$\mathcal{R}_V(\mathbf{a}_t^{\text{query}}) = \underset{v \in V}{\arg\max}^{k_V} \Big( \text{sim}(\phi(V_{\mathbf{a}_t^{\text{query}}}), \phi(v)) \Big), \quad \mathcal{F}_V^* = \bigcup_{v_i \in \mathcal{R}_V} \{(e_H, V_{e_H}) \mid v_i \in V_{e_H}, e_H \in E_H\}, \quad (7)$$

where $\phi(V_{\mathbf{a}_t^{\text{query}}})$ is the aggregated embedding of entities extracted from $\mathbf{a}_t^{\text{query}}$, $\phi(v)$ is the entity embedding, $k_V$ is the number of retrieved entities, and $V_{e_H}$ denotes the entity set of hyperedge $e_H$.

*(ii) Direct Hyperedge Retrieval.* In parallel, we directly retrieve hyperedges based on query-hyperedge similarity, and collect their associated relational facts:

$$\mathcal{R}_H(\mathbf{a}_t^{\text{query}}) = \underset{e_H \in E_H}{\arg\max}^{k_H} \big( \text{sim}(\phi(\mathbf{a}_t^{\text{query}}), \phi(e_H)) \big), \quad \mathcal{F}_H^* = \bigcup_{e_i \in \mathcal{R}_H} \{(e_i, V_{e_i}) \mid V_{e_i} \subseteq V\}, \quad (8)$$

where $\phi(\mathbf{a}_t^{\text{query}})$ is the query embedding, $\phi(e_H)$ is the hyperedge embedding, $k_H$ is the number of retrieved hyperedges, and $V_{e_i}$ denotes the entity set of hyperedge $e_i$.

*(iii) Fusion via Reciprocal Rank Aggregation.* To produce the final knowledge set, we merge results from both retrieval paths using reciprocal rank aggregation over hyperedges:

$$\mathbf{a}_t^{\text{ret}} = \mathcal{F}_{\mathbf{a}_t^{\text{query}}}^* = \text{Top-}k \left( \mathcal{F}_V^* \cup \mathcal{F}_H^*, \text{RankScore}(f) = \frac{1}{r_V} + \frac{1}{r_H} \right)_{\mathbf{a}_t^{\text{query}}}, \quad (9)$$

where $r_V$ and $r_H$ are the ranks of n-ary relational fact $f$ in $\mathcal{F}_V^*$ and $\mathcal{F}_H^*$ respectively (set to $\infty$ if absent), and $k$ is the number of retrieved facts $\mathbf{a}_t^{\text{ret}}$ returned to the agent.

**Optimization Objective for Agent Trajectories.** The agent aims to learn a reasoning trajectory $\tau \in \mathcal{T}_q$ that yields a faithful and contextually grounded answer $y_q$. Each trajectory $\tau = ((\mathbf{s}_1, \mathbf{a}_1), (\mathbf{s}_2, \mathbf{a}_2), \ldots, (\mathbf{s}_T, \mathbf{a}_T))$ comprises a sequence of actions executed over $\mathcal{G}_H$, defined as:

$$\max_{\theta} \quad \mathbb{E}_{\tau \sim \pi_\theta(\mathcal{T}_q | q; \mathcal{G}_H)} \left[ \log P(y_q \mid \tau) \right], \quad (10)$$

where $P(y_q \mid \tau)$ denotes the likelihood of the correct answer $y_q \sim \mathbf{a}_t^{\text{ans}}$ under trajectory $\tau$, guiding $\pi_\theta$ toward answer-consistent reasoning.

**Proposition 2.** *Multi-turn interaction with the graph environment improves retrieval efficiency.*

*Proof.* We provide experimental results in Section 5.5 and theoretical proofs in Appendix B.2. $\square$

### 4.3 OUTCOME-DIRECTED END-TO-END REINFORCEMENT LEARNING

To optimize the reasoning policy $\pi_\theta$ toward generating faithful and well-structured answers, we adopt an end-to-end reinforcement learning objective based on Group Relative Policy Optimization (GRPO) (Shao et al., 2024) $\mathcal{J}_{\text{GRPO}}(\theta)$ and design an outcome-directed reward function $R(\tau)$.

**End-to-end RL Objective $\mathcal{J}_{\text{GRPO}}(\theta)$.** Given a dataset question $q \in \mathcal{D}_Q$, the agent interacts with the knowledge hypergraph $\mathcal{G}_H$ to generate a group of multi-turn reasoning trajectories $\{\tau_i\}_{i=1}^N \subseteq \mathcal{T}_q$,

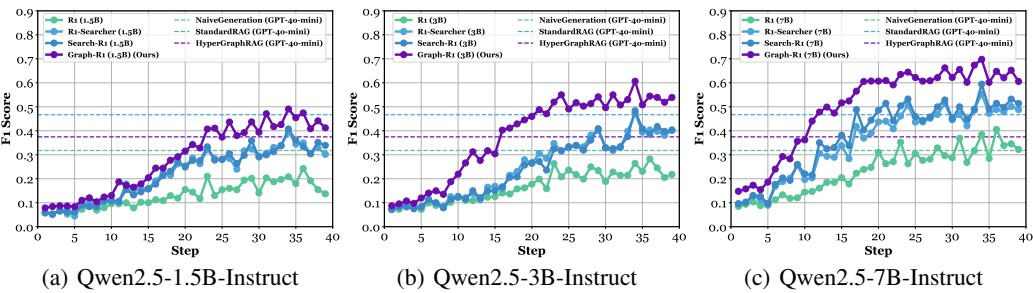

Figure 4: Step-wise F1 score on HotpotQA based on Qwen2.5 (1.5B, 3B, 7B), where Graph-R1 outperforms baselines and GPT-4o-mini variants (NaiveGeneration, StandardRAG, HyperGraphRAG).

where each $\tau_i = ((\mathbf{s}_1^{(i)}, \mathbf{a}_1^{(i)}), \ldots, (\mathbf{s}_T^{(i)}, \mathbf{a}_T^{(i)}))$ denotes a sequence of state-action pairs sampled from the environment. We optimize the policy $\pi_\theta$ using the GRPO-based objective, which is defined as:

$$\mathcal{J}_{\text{GRPO}}(\theta) = \mathbb{E}_{\left[\mathbf{s}_1 \sim \{\mathbb{P}(q)|q \in \mathcal{D}_Q\}, \{\tau_i\}_{i=1}^N \sim \pi_{\theta_{\text{old}}}(\mathcal{T}_q|\mathbf{s}_1; \mathcal{G}_H)\right]}$$

$$\left[ \frac{1}{N} \sum_{i=1}^N \frac{1}{|\tau_i|} \sum_{t=1}^{|\tau_i|} \min\left(\rho_\theta(\mathbf{a}_t^{(i)})\hat{A}(\tau_i), \text{clip}\left(\rho_\theta(\mathbf{a}_t^{(i)}), 1 \pm \epsilon\right)\hat{A}(\tau_i)\right) - \beta\,\mathbb{D}_{\text{KL}}(\pi_\theta\|\pi_{\text{ref}}) \right], \quad (11)$$

where $\rho_\theta(\mathbf{a}_t^{(i)}) = \dfrac{\pi_\theta(\mathbf{a}_t^{(i)} \mid \mathbf{s}_{t-1}^{(i)}; \mathcal{G}_H))}{\pi_{\theta_{\text{old}}}(\mathbf{a}_t^{(i)} \mid \mathbf{s}_{t-1}^{(i)}; \mathcal{G}_H))}$, and $\hat{A}(\tau_i) = \dfrac{R(\tau_i) - \text{mean}\left(\{R(\tau_j)\}_{j=1}^N\right)}{F_{\text{norm}}\left(\{R(\tau_j)\}_{j=1}^N\right)}$. $\quad$ (12)

Here, $\pi_\theta$ is the current policy, and $\pi_{\theta_{\text{old}}}$ is the behavior policy used for sampling. The importance ratio $\rho_\theta(\mathbf{a}_t^{(i)})$ adjusts for distribution shift, while the advantage $\hat{A}(\tau_i)$ normalizes the reward using a scaling function $F_{\text{norm}}(\cdot)$ (e.g., standard deviation). The clip$(\cdot)$ operator stabilizes updates by constraining policy shifts. A KL term $\mathbb{D}_{\text{KL}}(\pi_\theta \| \pi_{\text{ref}})$ regularizes toward a reference policy $\pi_{\text{ref}}$, with $\beta$ controlling its strength. This objective encourages high-reward, stable reasoning over $\mathcal{G}_H$.

**Outcome-directed Reward Function $R(\tau)$.** To meet outcome requirements, we define a reward function $R(\tau)$ composed of two parts: a *format reward* $R_{\text{format}}(\tau)$ and an *answer reward* $R_{\text{answer}}(\mathbf{a}_T^{\text{ans}})$, promoting both thoughtful retrieval and accurate answer generation.

*(i) Format Reward.* The format reward $R_{\text{format}}(\tau)$ encourages the agent to follow the intended reasoning structure. At each step $(\mathbf{s}_t, \mathbf{a}_t)$, we check whether the output includes a well-formed block $(\mathbf{a}_t^{\text{think}}, \alpha_t, \mathbf{a}_t^{\text{out}})$. Each valid step receives 0.5 reward, capped at 1.0 overall:

$$R_{\text{format}}(\tau) = \min\left(1.0,\ 0.5 \cdot \sum_{t=1}^T \mathbb{I}\left\{(\mathbf{a}_t^{\text{think}}, \alpha_t, \mathbf{a}_t^{\text{out}}) \text{ is well-formed}\right\}\right), \quad (13)$$

where $\mathbb{I}\{\cdot\}$ is an indicator function that returns 1 if the step output matches the expected format.

*(ii) Answer Reward.* The answer reward $R_{\text{answer}}(\mathbf{a}_T^{\text{ans}})$ measures the semantic correctness of the generated answer $\mathbf{a}_T^{\text{ans}}$ by comparing it with the ground-truth answer $y_q^*$ using a token-level F1 score:

$$R_{\text{answer}}(\mathbf{a}_T^{\text{ans}}) = \frac{2 \cdot |\text{tokens}(\mathbf{a}_T^{\text{ans}}) \cap \text{tokens}(y_q^*)|}{|\text{tokens}(\mathbf{a}_T^{\text{ans}})| + |\text{tokens}(y_q^*)|}, \quad (14)$$

where $|\cdot|$ denotes multiset cardinality. The function tokens$(\cdot)$ applies standard preprocessing including lowercasing and whitespace-based tokenization.

*(iii) Overall Outcome Reward.* The total reward for a reasoning trajectory $\tau$ is defined as:

$$R(\tau) = -1.0 + R_{\text{format}}(\tau) + \mathbb{I}\{R_{\text{format}}(\tau) = 1.0\} \cdot R_{\text{answer}}(\mathbf{a}_T^{\text{ans}}), \text{ where } \mathbf{a}_T^{\text{ans}} \in \tau, \quad (15)$$

ensuring that answer correctness is only rewarded when the format is structurally valid. With the outcome-directed reward $R(\tau)$, high answer quality $\mathbf{a}_T^{\text{ans}}$ is attainable through structurally coherent and reasoning-complete trajectories $\tau$ with multi-turn iteration with knowledge hypergraph $\mathcal{G}_H$.

**Proposition 3.** *End-to-end RL bridges the gap between graph-based knowledge and language.*

*Proof.* We provide experimental results in Section 5.6 and theoretical proofs in Appendix B.3. □

| Method | 2Wiki. | | HotpotQA | | Musique | | NQ | | PopQA | | TriviaQA | | Avg. | | | |
|---|---|---|---|---|---|---|---|---|---|---|---|---|---|---|---|---|
| | F1 | G-E | F1 | G-E | F1 | G-E | F1 | G-E | F1 | G-E | F1 | G-E | EM | F1 | R-S | G-E |
| *GPT-4o-mini* | | | | | | | | | | | | | | | | |
| ⊙⊘ NaiveGeneration | 17.03 | 74.86 | 31.79 | 78.48 | 11.45 | 76.61 | 21.59 | 84.64 | 25.95 | **72.75** | 47.73 | 83.33 | 11.36 | 25.92 | - | 78.45 |
| ⊙⊞ StandardRAG | **22.31** | 73.02 | **46.70** | **81.88** | 17.31 | 74.93 | **26.85** | 84.55 | **30.58** | 69.42 | **48.55** | 84.63 | **18.10** | **32.05** | 52.68 | 78.07 |
| ⊙⋏ GraphRAG | 16.02 | 72.81 | 31.67 | 77.37 | 15.14 | 74.43 | 20.31 | 82.36 | 20.92 | 65.88 | 45.13 | 82.76 | 12.50 | 24.87 | | 75.94 |
| ⊙⋏ LightRAG | 16.59 | 71.94 | 30.70 | 73.42 | 14.39 | 73.75 | 19.09 | 80.20 | 20.47 | 67.76 | 40.18 | 81.60 | 9.77 | 23.57 | 47.42 | 74.78 |
| ⊙⋏ PathRAG | 12.42 | 67.19 | 23.12 | 71.81 | 11.49 | 69.94 | 20.01 | 81.99 | 15.65 | 60.58 | 37.44 | 80.94 | 7.03 | 20.02 | 46.71 | 72.08 |
| ⊙⋏ HippoRAG2 | 16.27 | 68.78 | 31.78 | 76.43 | 12.37 | 73.05 | 24.56 | **84.65** | 21.10 | 63.31 | 46.86 | 83.55 | 13.80 | 25.49 | 36.41 | 74.96 |
| ⊙⋏ HyperGraphRAG | 21.14 | **76.76** | 37.46 | 80.50 | **20.40** | **79.29** | 22.95 | 81.22 | 29.48 | 70.55 | 44.95 | **85.20** | 13.15 | 29.40 | **61.82** | 78.92 |
| *Qwen2.5-1.5B-Instruct* | | | | | | | | | | | | | | | | |
| ⊙⊘ NaiveGeneration | 7.78 | 49.13 | 4.27 | 45.77 | 2.35 | 46.63 | 6.03 | 46.74 | 10.06 | 42.67 | 8.10 | 52.92 | 1.17 | 6.43 | - | 47.31 |
| ⊙⊞ StandardRAG | 11.46 | 55.38 | 9.93 | 52.91 | 3.18 | 39.46 | 11.39 | 59.73 | 13.08 | 50.29 | 17.43 | 60.52 | 5.73 | 11.08 | 52.84 | 53.05 |
| ◖⊘ SFT | 13.26 | 34.72 | 13.61 | 38.93 | 5.14 | 28.50 | 11.56 | 46.61 | 15.61 | 31.35 | 26.18 | 46.66 | 9.83 | 14.23 | - | 37.80 |
| ◖⊘ R1 | 26.28 | 47.48 | 20.07 | 44.43 | 4.84 | 39.12 | 16.75 | 45.95 | 21.36 | 44.50 | 34.78 | 48.59 | 14.19 | 20.68 | - | 45.01 |
| ◖⊞ Search-R1 | 28.43 | 60.61 | 39.99 | 64.16 | 4.69 | 39.32 | 20.26 | 59.93 | 39.63 | 58.19 | 44.16 | 63.01 | 23.18 | 29.53 | 50.45 | 57.54 |
| ◖⊞ R1-Searcher | 28.01 | 58.81 | **41.50** | 61.54 | 6.26 | 38.31 | **36.86** | **60.79** | 38.37 | 56.02 | 42.57 | 61.24 | 23.70 | 32.26 | 50.68 | 56.12 |
| ◖⋏ Graph-R1 (ours) | **35.13** | **65.73** | 40.62 | **65.30** | **28.28** | **58.82** | 35.62 | 59.13 | **43.55** | **66.46** | **57.36** | **70.83** | **31.90** | **40.09** | **59.35** | **64.38** |
| *Qwen2.5-3B-Instruct* | | | | | | | | | | | | | | | | |
| ⊙⊘ NaiveGeneration | 7.59 | 55.00 | 11.16 | 53.75 | 3.67 | 54.00 | 8.90 | 57.18 | 10.89 | 49.08 | 10.89 | 48.16 | 3.26 | 8.85 | - | 52.86 |
| ⊙⊞ StandardRAG | 12.52 | 60.01 | 15.41 | 62.51 | 2.92 | 50.40 | 10.69 | 65.13 | 14.70 | 57.25 | 21.92 | 68.43 | 3.39 | 13.03 | 52.69 | 60.62 |
| ◖⊘ SFT | 12.40 | 52.31 | 16.48 | 51.35 | 5.04 | 51.31 | 11.23 | 58.20 | 16.95 | 46.42 | 33.02 | 59.98 | 9.64 | 15.85 | - | 53.26 |
| ◖⊘ R1 | 28.45 | 56.92 | 25.33 | 55.38 | 8.07 | 47.53 | 21.51 | 55.11 | 27.11 | 48.65 | 47.91 | 60.74 | 19.66 | 26.40 | - | 54.06 |
| ◖⊞ Search-R1 | 38.04 | 54.39 | 43.84 | 69.32 | 7.65 | 46.43 | 37.96 | 52.90 | 38.67 | 63.74 | 47.99 | 60.37 | 28.65 | 35.69 | 49.99 | 57.86 |
| ◖⊞ R1-Searcher | 23.50 | 55.86 | 42.44 | 64.60 | 12.81 | 50.07 | 36.53 | 63.33 | 40.18 | 66.23 | 54.00 | 60.52 | 27.08 | 34.91 | 49.98 | 60.10 |
| ◖⋏ Graph-R1 (ours) | **57.56** | **76.45** | **56.75** | **77.46** | **40.51** | **67.84** | **44.75** | **69.92** | **45.65** | **71.27** | **62.31** | **75.01** | **42.45** | **51.26** | **60.19** | **72.99** |
| *Qwen2.5-7B-Instruct* | | | | | | | | | | | | | | | | |
| ⊙⊘ NaiveGeneration | 12.25 | 66.75 | 16.58 | 65.31 | 4.06 | 65.47 | 13.00 | 69.56 | 12.82 | 60.50 | 24.51 | 72.65 | 3.12 | 13.87 | - | 66.71 |
| ⊙⊞ StandardRAG | 12.75 | 60.06 | 21.10 | 66.13 | 4.53 | 59.84 | 15.97 | 70.49 | 16.10 | 60.86 | 24.90 | 73.71 | 5.34 | 15.89 | 52.67 | 65.18 |
| ◖⊘ SFT | 20.28 | 63.85 | 27.59 | 65.65 | 10.02 | 63.50 | 19.02 | 68.19 | 27.93 | 56.31 | 39.21 | 70.25 | 15.57 | 24.01 | - | 64.63 |
| ◖⊘ R1 | 30.99 | 59.19 | 37.05 | 60.12 | 14.53 | 49.39 | 28.45 | 57.63 | 30.35 | 53.38 | 57.33 | 66.73 | 25.91 | 33.12 | - | 57.74 |
| ◖⊞ Search-R1 | 41.29 | 70.26 | 50.85 | 73.85 | 22.35 | 57.68 | 45.88 | 67.58 | 50.76 | 66.98 | 65.98 | 76.15 | 38.54 | 46.19 | 51.60 | 68.60 |
| ◖⊞ R1-Searcher | 33.96 | 69.61 | 46.36 | 74.56 | 16.63 | 59.05 | 44.93 | 68.54 | 47.12 | 66.74 | 64.76 | 75.95 | 34.51 | 42.29 | 51.26 | 69.08 |
| ◖⋏ Graph-R1 (ours) | **65.04** | **82.42** | **62.69** | **80.03** | **46.17** | **71.42** | **49.87** | **70.97** | **51.22** | **73.43** | **71.93** | **79.11** | **48.57** | **57.82** | **60.40** | **76.23** |

Table 2: Main results with best in **bold**. ⊙ means prompt engineering, ◖ means training, ⊘ means no knowledge interaction, ⊞ means chunk-based knowledge, and ⋏ means graph-based knowledge.

# 5 EXPERIMENTS

This section presents the experimental setup, main results, and analysis. We answer the following research questions (RQs): **RQ1:** Does Graph-R1 outperform other methods? **RQ2:** Does the main component of Graph-R1 work, and how is its comparative analysis? **RQ3-6:** How are construction cost, retrieval efficiency, generation quality, and generalizability of Graph-R1, respectively?

## 5.1 EXPERIMENTAL SETUP

**Datasets.** To evaluate the performance of Graph-R1, we conduct experiments across six standard RAG datasets (Jin et al., 2025b): 2WikiMultiHopQA (**2Wiki.**) (Ho et al., 2020), **HotpotQA** (Yang et al., 2018), **Musique** (Trivedi et al., 2022), Natural Questions (**NQ**) (Kwiatkowski et al., 2019), **PopQA** (Mallen et al., 2023), and **TriviaQA** (Joshi et al., 2017). More details are in Appendix F.

**Baselines.** We mainly compare Graph-R1 with **NaiveGeneration**, **StandardRAG** (Lewis et al., 2020), **SFT** (Zheng et al., 2024), **R1** (Shao et al., 2024), **Search-R1** (Jin et al., 2025a), and **R1-Searcher** (Song et al., 2025) at three `Qwen2.5` (Qwen et al., 2025) scales: 1.5 B, 3 B, and 7 B. We also compare **GraphRAG** (Edge et al., 2025), **LightRAG** (Guo et al., 2025), **PathRAG** (Chen et al., 2025), **HippoRAG2** (Gutiérrez et al., 2025), and **HyperGraphRAG** (Luo et al., 2025a) based on `GPT-4o-mini` (OpenAI et al., 2024a) as a reference. More details are in Appendix G.

**Evaluation Metrics.** We evaluate Graph-R1 and baselines with four metrics: Exact Match (**EM**), **F1**, Retrieval Similarity (**R-S**), and Generation Evaluation (**G-E**). More details are in Appendix H.

**Implementation Details.** We use `GPT-4o-mini` for knowledge construction in Graph-R1 and GraphRAG baselines. For retrieval, we use `bge-large-en-v1.5` (Chen et al., 2023) in all variants. All experiments are done on 4 NVIDIA A100 GPUs (80GB). More details are in Appendix I.

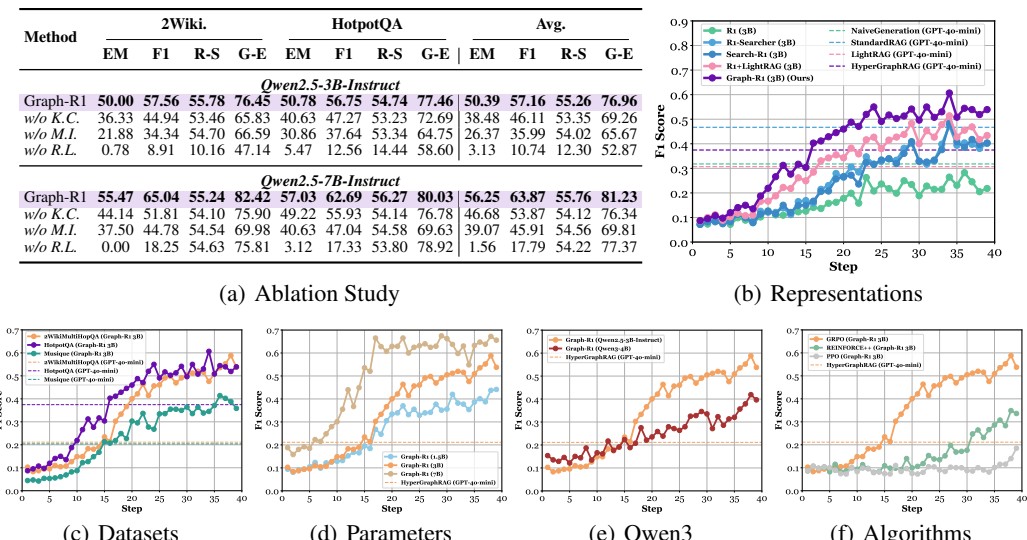

| Method | 2Wiki. | | | | HotpotQA | | | | Avg. | | | |
|---|---|---|---|---|---|---|---|---|---|---|---|---|
| | EM | F1 | R-S | G-E | EM | F1 | R-S | G-E | EM | F1 | R-S | G-E |
| *Qwen2.5-3B-Instruct* | | | | | | | | | | | | |
| Graph-R1 | **50.00** | **57.56** | **55.78** | **76.45** | **50.78** | **56.75** | **54.74** | **77.46** | **50.39** | **57.16** | **55.26** | **76.96** |
| *w/o K.C.* | 36.33 | 44.94 | 53.46 | 65.83 | 40.63 | 47.27 | 53.23 | 72.69 | 38.48 | 46.11 | 53.35 | 69.26 |
| *w/o M.I.* | 21.88 | 34.34 | 54.70 | 66.59 | 30.86 | 37.64 | 53.34 | 64.75 | 26.37 | 35.99 | 54.02 | 65.67 |
| *w/o R.L.* | 0.78 | 8.91 | 10.16 | 47.14 | 5.47 | 12.56 | 14.44 | 58.60 | 3.13 | 10.74 | 12.30 | 52.87 |
| *Qwen2.5-7B-Instruct* | | | | | | | | | | | | |
| Graph-R1 | **55.47** | **65.04** | **55.24** | **82.42** | **57.03** | **62.69** | **56.27** | **80.03** | **56.25** | **63.87** | **55.76** | **81.23** |
| *w/o K.C.* | 44.14 | 51.81 | 54.10 | 75.90 | 49.22 | 55.93 | 54.14 | 76.78 | 46.68 | 53.87 | 54.12 | 76.34 |
| *w/o M.I.* | 37.50 | 44.78 | 54.54 | 69.98 | 40.63 | 47.04 | 54.58 | 69.63 | 39.07 | 45.91 | 54.56 | 69.81 |
| *w/o R.L.* | 0.00 | 18.25 | 54.63 | 75.81 | 3.12 | 17.33 | 53.80 | 78.92 | 1.56 | 17.79 | 54.22 | 77.37 |

(a) Ablation Study        (b) Representations

(c) Datasets     (d) Parameters     (e) Qwen3     (f) Algorithms

Figure 5: (a) Ablation study of Graph-R1. (b-f) Performance comparison across different kinds of knowledge representations, RAG datasets, model parameters, Qwen versions, and RL algorithms.

## 5.2 MAIN RESULTS (RQ1)

As shown in Table 2, we compare Graph-R1 with baselines across different base models, and observe that Graph-R1 consistently outperforms all baselines. In addition, we have two key observations.

**RL Unlocks the Power of Graph Representations.** Prompt-only GraphRAG methods often underperform StandardRAG, showing that graph structures alone are not sufficient. Graph-R1, with multi-turn RL optimization, fully exploits structural signals, achieving 57.28 F1 under Qwen2.5-7B-Instruct, surpassing StandardRAG (32.05), HyperGraphRAG (29.40) and Search-R1 (46.19).

**Larger Base Model Further Enhances Performance.** As base model size increases from 1.5B to 3B and 7B, Graph-R1 achieves steadily higher F1 scores: 40.09, 51.26, and 57.82. Moreover, its gap over other RL-enhanced baselines such as Search-R1 and R1-Searcher becomes increasingly evident. This shows that larger models better exploit the synergy between graph structures and RL.

## 5.3 ABLATION STUDY AND COMPARATIVE ANALYSIS (RQ2)

As shown in Figures 5, we conduct an ablation study and comparative analysis on Graph-R1.

**Ablation Study.** We remove three core components of Graph-R1: knowledge construction (K.C.), multi-turn interaction (M.I.), and reinforcement learning (R.L.), to assess their individual contributions. As shown in Figure 5(a), removing any module leads to performance degradation.

**Comparison with Different Knowledge Representations.** As shown in Figures 4 and 5(b), models without external knowledge (green) perform the worst. Chunk-based knowledge with RL (blue) performs better, but is still inferior to graph-based methods using binary relations (pink), while hypergraph-based knowledge with RL (red) achieves the highest ceiling. This demonstrates that, when combined with RL, stronger knowledge representations yield higher performance potential.

**Comparison across Datasets and Base Models.** As shown in Figures 5(c) and 5(d), Graph-R1 consistently outperforms baselines across different datasets and parameter sizes, showcasing strong scalability. Interestingly, Figure 5(e) shows that when Graph-R1 is trained on Qwen3 (4B) (Yang et al., 2025a), which is already well trained by RL, the model tends to over-rely on its own internal reasoning. Despite a stronger starting point, its overall performance ceiling appears slightly lower.

**Comparison with Different RL Algorithms.** Figure 5(f) compares different RL strategies. GRPO significantly outperforms REINFORCE++ (Hu et al., 2025) and PPO (Schulman et al., 2017), achieving the highest F1. This confirms that GRPO facilitates more stable training and stronger multi-turn graph reasoning, making it a favorable choice for training agentic GraphRAG models.

## 5.4 ANALYSIS OF GRAPH-R1'S CONSTRUCTION COST (RQ3)

As shown in Table 3, we utilize metrics: time per 1K tokens (TP1KT), cost per 1M tokens (CP1MT), number of nodes & edges, time per query (TPQ), cost per 1K queries (CP1KQ), and final F1 score.

**Construction Cost.** Graph-R1 requires only 5.69 seconds and $2.81 per 1K tokens for knowledge construction, lower than GraphRAG (8.04s, $3.35) and HyperGraphRAG (6.76s, $4.14). Generating over 120K nodes and 98K edges, Graph-R1 maintains a semantically rich structure.

Table 3: Time & Cost Comparisons on 2Wiki.

| Method | Knowledge Construction | | | | Retrieval & Generation | | |
|---|---|---|---|---|---|---|---|
| | TP1KT | CP1MT | #Node | #Edge | TPQ | CP1KQ | F1 |
| NaiveGeneration | 0 s | 0 $ | - | - | 3.7 s | 0.16 $ | 17.0 |
| StandardRAG | 0 s | 0 $ | - | - | 4.1 s | 1.35 $ | 22.3 |
| GraphRAG | 8.04 s | 3.35 $ | 7,771 | 4,863 | 7.4 s | 3.97 $ | 16.0 |
| LightRAG | 6.84 s | 4.07 $ | 59,197 | 24,596 | 12.2 s | 8.11 $ | 16.6 |
| PathRAG | 6.84 s | 4.07 $ | 59,197 | 24,596 | 15.8 s | 8.28 $ | 12.4 |
| HippoRAG2 | 3.25 s | 1.26 $ | 11,819 | 40,654 | 8.8 s | 7.68 $ | 16.3 |
| HyperGraphRAG | 6.76 s | 4.14 $ | 173,575 | 114,426 | 9.6 s | 8.76 $ | 21.1 |
| Graph-R1 (7B) (ours) | 5.69 s | 2.81 $ | 120,499 | 98,073 | 7.0 s | 0 $ | 65.0 |

**Generation Cost.** By leveraging end-to-end RL and localized knowledge retrieval, Graph-R1 achieves not only the best F1 but also a response time of 7.0s per query and a generation cost of $0, outperforming baselines such as HyperGraphRAG (9.6s, $8.76), highlighting its superior potential for real-world deployment.

## 5.5 ANALYSIS OF GRAPH-R1'S RETRIEVAL EFFICIENCY (RQ4)

As shown in Figure 6, to evaluate Graph-R1's retrieval efficiency, we analyze it from (a) response length, (b) number of interaction turns, and (c) performance with average retrieval content lengths.

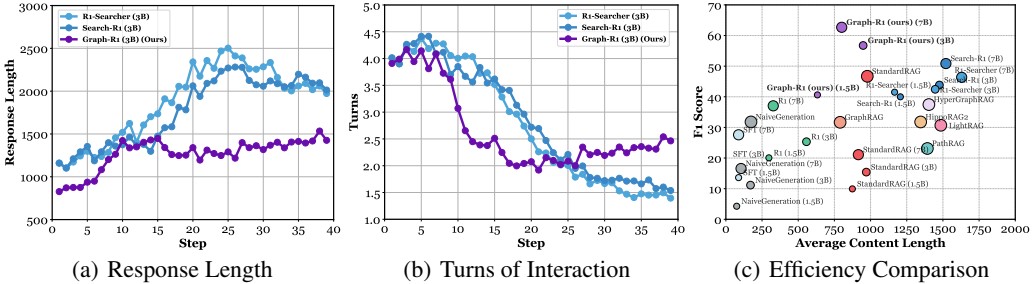

| (a) Response Length | (b) Turns of Interaction | (c) Efficiency Comparison |
|---|---|---|

Figure 6: Step-wise response length & turns of interaction, and efficiency comparison on HotpotQA.

**Tendency toward More Concise Thinking and Adequate Interaction.** As shown in Figures 6(a) and 6(b), Graph-R1 generates shorter responses and conducts more interaction turns, averaging around 1200-1500 tokens and 2.3-2.5 turns, leading to more stable and accurate retrieval.

**Balancing Performance and Retrieved Content Length.** As shown in Figure 6(c), Graph-R1 achieves the highest F1 scores with a moderate amount of average retrieved content compared to other methods, balancing input length and performance through its multi-turn interaction strategy.

## 5.6 ANALYSIS OF GRAPH-R1'S GENERATION QUALITY (RQ5)

As shown in Figure 7, we evaluate the generation quality in seven dimensions and present a case study in Table 4.

**High-Quality Generation Performance.** Graph-R1 outperforms all RL-based baselines and achieves generation quality comparable to GPT-4o-mini-based methods like HyperGraphRAG, with strong results in Correctness (86.9), Relevance (95.2), and Logical Coherence (88.5).

**RL Bridges the Gap Between Graph & Language.** HyperGraphRAG performs similarly to StandardRAG, indicating limited gains from graph structure alone. In contrast, Graph-R1 achieves a much higher Overall score (82.4 vs. 70.3) than Search-R1, showing that graph-based reasoning becomes truly effective when combined with RL.

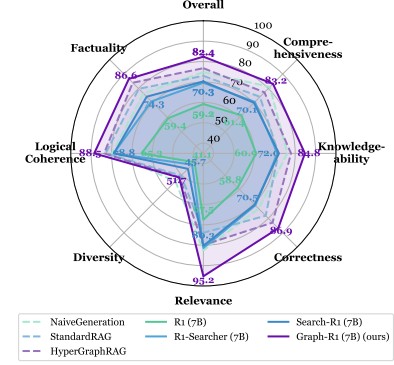

Figure 7: Generation Evaluations.

| Query | When was the director of film Ingmar'S Inheritance born? | | | | | | | | |
|---|---|---|---|---|---|---|---|---|---|
| Golden Answers | ['18 November 1888'] | | | | | | | | |
| **GPT-4o-mini** | **NaiveGeneration** | | | **StandardRAG** | | | **HyperGraphRAG** | | |
| Generation | `<think>` \nTo answer this question, I first need to identify who the director of "Ingmar's Inheritance" is. The film "Ingmar's Inheritance" is directed by a notable filmmaker. Researching the film, it turns out that it was directed by the Swedish director, "Ingmar Bergman." I need to check the birth date of Ingmar Bergman to provide the correct answer. Ingmar Bergman was born on July 14, 1918. Since the question is specifically asking about the birth of the director of the film rather than the film's release date or other details, I will now formulate the answer based on this information.\n `</think>` \n `<answer>` \nIngmar Bergman was born on July 14, 1918.\n `</answer>` | | | `<think>` \nThe question asks for the birth year of the director of the film "Ingmar\'s Inheritance." However, the provided knowledge does not include information about the film "Ingmar\'s Inheritance" or its director..."\n `</think>` \n `<answer>` \nThe information necessary to answer the question about the director of "Ingmar\'s Inheritance" is not available in the provided knowledge. Therefore, I cannot provide a specific birth date for that director. However, Bille August, a noted director with connections to Ingmar Bergman, was born on November 9, 1948. It's unclear if he is associated with "Ingmar\'s Inheritance." \n `</answer>` | | | `<think>` \nTo answer the question, I need to identify the director of the film "Ingmar\'s Inheritance." The information available mentions that "Ingmar\'s Inheritance" is a silent drama film directed by Gustaf Molander. The provided data does not specifically give a birthdate for Gustaf Molander. However, I find that the knowledge base also contains information about Ernst Ingmar Bergman, a prominent director, with dates of birth and death, but he is not related to "Ingmar\'s Inheritance." Since only Gustaf Molander is mentioned as the director of that specific film, and I don't have his birthdate recorded in the provided knowledge, I cannot provide an answer to this question.\n `</think>` `<answer>` \nThe birthdate of Gustaf Molander, the director of "Ingmar\'s Inheritance", is not provided in the knowledge base, so I cannot answer the question. \n `</answer>` | | |
| Evaluation Score | F1 | R-S | G-E | F1 | R-S | G-E | F1 | R-S | G-E |
| | 0.00 | - | 55.71 | 3.70 | 39.52 | 55.71 | 0.00 | 38.93 | 60.00 |
| **Qwen2.5-7B-Instruct** | **R1 (7B)** | | | **Search-R1 (7B)** | | | **Graph-R1 (7B)** | | |
| Generation | `<think>` \nTo answer when the director of the film "Ingmar\'s Inheritance" was born, I first need to find out who the director of this film is.\nThe film "Ingmar\'s Inheritance" was directed by Garth Davis. Now I need to find out when Garth Davis was born.\nGarth Davis was born on December 26, 1970. Therefore, the director of the film "Ingmar\'s Inheritance" was born on December 26, 1970.\n `</think>` \n `<answer>` \nDecember 26, 1970 \n `</answer>` | | | `<think>` \nTo answer this question, I need to find the director of the film "Ingmar\'s Inheritance" and then determine their birth date.\n `</think>` \n `<query>` director of Ingmar\'s Inheritance `</query>` \n `<knowledge>` \n{"results": ["Weizscker studied at the University of Tcbingen. While serving in the administration of King William II of Wcrttemberg, he was raised to personal nobility on 24 February 1897...",...]}\n `</knowledge>` \n `<think>` \nFrom the information provided, Baldvin Zophon íasson is the director of "Ingmar\'s Inheritance" and he was born in 1978 in Akureyri.\n `</think>` \n `<answer>` \n1978 `</answer>` | | | `<think>` \nTo answer this question, I need to find out the director of the film "Ingmar\'s Inheritance" and then determine the director\'s birth year. Let\'s start by finding the director of the film.\n `</think>` \n `<query>` director of Ingmar\'s Inheritance `</query>` \n `<knowledge>` \n{"results": [{"<knowledge>": "\\"Ernst Ingmar Bergman (14 July 1918 \u2013 30 July 2007) was a Swedish director, writer, and producer who worked in film, television, theatre and radio.\\"", "<coherence>": 1.25}, {...}, ...]}\n `</knowledge>` \n `<think>` \nFrom the knowledge query, it seems that "Ingmar\'s Inheritance" was directed by Gustaf Molander. Now, I need to find out Gustaf Molander\'s birth year to answer the question.\n `</think>` \n `<query>` Gustaf Molander birth year `</query>` \n `<knowledge>` \n{"results": [{"<knowledge>": "\\"Gustaf Molander was born in Helsingfors (now Helsinki) in the Grand Duchy of Finland (in the Russian Empire), where his father was working at the Swedish Theatre.\\"", "<coherence>": 1.5}, {...}, ...]}\n `</knowledge>` \n `<think>` \nFrom the knowledge query, we found that Gustaf Molander was born on November 18, 1888. This is the director of the film "Ingmar\'s Inheritance".\n `</think>` \n `<answer>` \nThe director of the film "Ingmar\'s Inheritance", Gustaf Molander, was born on November 18, 1888.\n `</answer>` | | |
| Evaluation Score | F1 | R-S | G-E | F1 | R-S | G-E | F1 | R-S | G-E |
| | 0.00 | - | 41.43 | 0.00 | 40.02 | 64.29 | 37.50 | 45.83 | 88.57 |

Table 4: Case study on generation quality under a query, comparing NaiveGeneration, Standard-RAG, HyperGraphRAG based on GPT-4o-mini, with R1 (7B), Search-R1 (7B), and Graph-R1 (7B).

**Case Study Analysis.** As shown in Table 4, NaiveGeneration and R1 fail to provide the correct answer, and both StandardRAG and HyperGraphRAG also fall short despite using structured prompts. Search-R1, though RL-enhanced, shows limited improvement due to weak retrieval grounding. In contrast, Graph-R1 accurately identifies both the director and birthdate, achieves the highest G-E score (88.57), and demonstrates that RL is more effective with graph-based knowledge interaction.

## 5.7 ANALYSIS OF GRAPH-R1'S GENERALIZABILITY ON O.O.D. SETTINGS (RQ6)

As shown in Figure 8, to verify generalization, we conduct O.O.D. cross-validation for Search-R1 (3B) & Graph-R1 (3B) across six datasets: (a-b) F1 comparison, and (c-d) O.O.D.-to-I.I.D. ratios.

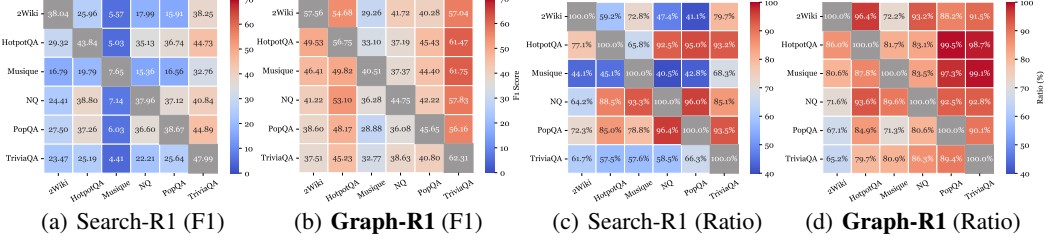

(a) Search-R1 (F1)  (b) **Graph-R1** (F1)  (c) Search-R1 (Ratio)  (d) **Graph-R1** (Ratio)

Figure 8: F1 comparison and performance ratios across six datasets under O.O.D. cross-validation.

**F1 Performance Across Datasets.** Figures 8(a) and 8(b) show that Graph-R1 outperforms Search-R1 on six datasets in O.O.D. validation, with notable gains on NQ and TriviaQA. Its multi-turn interaction with hypergraph retrieval ensures more stable performance under distribution shifts.

**Robust Generalization Ability.** Figures 8(c) and 8(d) show that Graph-R1 achieves higher O.O.D.-to-I.I.D. ratios than Search-R1, often above 85% and exceeding 90% in some cases, reflecting its strong robustness and cross-domain generalizability via end-to-end RL over knowledge hypergraph.

## 6 CONCLUSION

In this work, we introduce Graph-R1, an agentic GraphRAG framework powered by end-to-end RL. By introducing lightweight knowledge hypergraph construction and modeling retrieval as a multi-turn interaction process, Graph-R1 bridges graph-structured knowledge with natural language generation. A unified reward mechanism enables outcome-directed reasoning that outperforms prior GraphRAG methods and RL-enhanced baselines. Experiments across six benchmarks demonstrate Graph-R1's superiority in accuracy, retrieval efficiency, generation quality, and generalizability.

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

APPENDIX

# A PROMPTS USED IN GRAPH-R1

## A.1 KNOWLEDGE HYPERGRAPH CONSTRUCTION PROMPT

As shown in Figure 9, we use the same n-ary relation-extraction prompt as HyperGraphRAG (Luo et al., 2025a). Moreover, we streamline knowledge-hypergraph construction by skipping confidence-score calculations and adopting a simpler semantic-retrieval method, reducing construction costs while maintaining equivalent knowledge-representation.

```
-Goal-
Given a text document that is potentially relevant to this activity and a list of entity types, identify all entities of those types from the text and all relationships among the identified entities.
Use {language} as output language.

-Steps-
1. Divide the text into several complete knowledge segments. For each knowledge segment, extract the following information:
-- knowledge_segment: A sentence that describes the context of the knowledge segment.
Format each knowledge segment as ("hyper-relation"{tuple_delimiter}<knowledge_segment>)

2. Identify all entities in each knowledge segment. For each identified entity, extract the following information:
- entity_name: Name of the entity, use same language as input text. If English, capitalized the name.
- entity_type: Type of the entity.
- entity_description: Comprehensive description of the entity's attributes and activities.
Format each entity as ("entity"{tuple_delimiter}<entity_name>{tuple_delimiter}<entity_type>{tuple_delimiter}<entity_description>)

3. Return output in {language} as a single list of all the entities and relationships identified in steps 1 and 2. Use **{record_delimiter}** as the list delimiter.

4. When finished, output {completion_delimiter}

######################
-Examples-
######################
{examples}

######################
-Real Data-
######################
Text: {input_text}
######################
```

Figure 9: Prompt for n-ary relation extraction $\pi_{\text{ext}}$ in Equation 4.

## A.2 AGENTIC KNOWLEDGE REASONING PROMPT

The initial knowledge-reasoning prompt has been shown in Table 1. Through several rounds of interaction with the environment, the Graph-R1 agent keeps adding information, readying the prompt for the next exchange. As shown in Figure 10, we present a case of the final prompt produced by the full agentic knowledge-reasoning process.

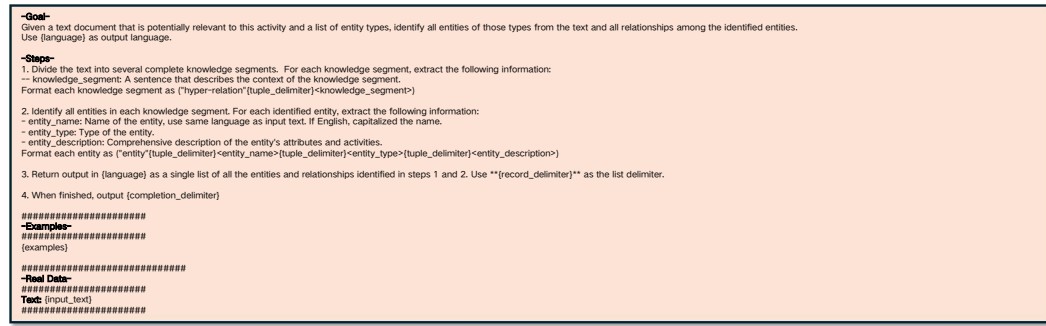

Figure 10: Prompt for agentic knowledge reasoning $\pi_\theta$ in Equation 6.

# B  THEORETICAL PROOF

## B.1  PROOF OF PROPOSITION 1

**Proposition 1.** *Graph-structured knowledge boosts agent accuracy by richer representation.*

*Proof.* Let the knowledge base $\mathcal{K}$ be encoded into two forms: a graph $R_G$ and a linear chunk set $R_C$, where $R_C = g(R_G)$ is a deterministic transformation that discards edge information. For a query $Q$ and ground-truth answer $A^\star$, the agent's internal belief at step $t$ is $h_t$. Each step performs retrieval $\mathcal{E}_t(R)$ and Bayesian update, forming the recurrence:

$$h_{t+1} = f(h_t, R). \tag{16}$$

Define the Lyapunov function as $V_R(h_t) = -\log P(A^\star \mid h_t)$, measuring how far the agent is from certainty. Its update is:

$$\Delta V_R(h_t) = -\log \frac{P(\mathcal{E}_t(R) \mid A^\star)}{\sum_a P(a \mid h_t) P(\mathcal{E}_t(R) \mid a)}. \tag{17}$$

Graphs can capture more relevant facts in shorter contexts due to explicit edges, leading to higher information density $\delta_R$ and more negative $\Delta V_R(h_t)$ in expectation. Thus, $V_R(h_t)$ decreases faster with graphs, indicating faster convergence. From an information-theoretic view, the mutual information evolves as:

$$I(A^\star; h_{t+1} \mid Q) = I(A^\star; h_t \mid Q) + I(A^\star; \mathcal{E}_t(R) \mid h_t, Q), \tag{18}$$

and since graphs provide denser evidence, we have $I_{R_G}(A^\star; h_T \mid Q) \geq I_{R_C}(A^\star; h_T \mid Q)$. Then by Fano's inequality,

$$P_e(R) \leq \frac{H(A^\star \mid Q) - I_R(A^\star; h_T \mid Q) + 1}{\log |\mathcal{A}|}, \tag{19}$$

which implies $P_e(R_G) \leq P_e(R_C)$, i.e., $\mathrm{Acc}(R_G) \geq \mathrm{Acc}(R_C)$, with strict inequality when the graph contains structural relations not recoverable from text.

In summary, the graph-structured representation offers higher information density per retrieval, accelerates belief convergence via Lyapunov descent, and accumulates more mutual information, leading to provably higher answer accuracy. $\square$

## B.2  PROOF OF PROPOSITION 2

**Proposition 2.** *Multi-turn interaction with the graph environment improves retrieval efficiency.*

*Proof.* Let the graph-structured knowledge base be denoted by $R_G$, and let $A^\star$ be the ground-truth answer. Suppose the retrieval cost is measured by the number of tokens retrieved, and we fix a total budget of $B$ tokens. A single-turn retrieval strategy selects a fixed token set $\mathcal{E}_{\mathrm{static}}$ of size $B$, independent of any intermediate reasoning. This leads to a posterior belief $P(A^\star \mid Q, \mathcal{E}_{\mathrm{static}})$ and yields total information gain

$$I_{\mathrm{static}} = I(A^\star; \mathcal{E}_{\mathrm{static}} \mid Q) = H(A^\star \mid Q) - H(A^\star \mid Q, \mathcal{E}_{\mathrm{static}}), \tag{20}$$

where $Q$ is the query and $H(\cdot)$ denotes entropy. In contrast, an adaptive multi-turn strategy $\pi$ divides the budget across $T$ rounds as $B = \sum_{t=1}^{T} B_t$. At each round $t$, the agent uses prior evidence $\mathcal{H}_{t-1} = \{\mathcal{E}_1, \ldots, \mathcal{E}_{t-1}\}$ to update its internal belief $h_{t-1}$ and selects new evidence $\mathcal{E}_t$ of size $B_t$ by actively exploring the graph based on current uncertainty. The updated belief $h_t$ is obtained via Bayesian inference, and the entire process forms a dynamic system:

$$h_t = f(h_{t-1}, \mathcal{E}_t, R_G). \tag{21}$$

To evaluate retrieval progress, we define a Lyapunov-style potential function $V_t = H(A^\star \mid Q, \mathcal{H}_t)$, which quantifies the remaining uncertainty after round $t$. Each retrieval step reduces entropy by:

$$V_{t-1} - V_t = I(A^\star; \mathcal{E}_t \mid Q, \mathcal{H}_{t-1}), \tag{22}$$

which is precisely the mutual information contributed by $\mathcal{E}_t$ given past evidence. Let $\rho_t$ denote the information gain per token at round $t$:

$$\rho_t = \frac{I(A^\star; \mathcal{E}_t \mid Q, \mathcal{H}_{t-1})}{B_t}, \tag{23}$$

and define the average information density of the static strategy as:

$$\rho_{\text{static}} = \frac{I_{\text{static}}}{B}.$$ (24)

Since the adaptive agent can tailor each $\mathcal{E}_t$ to the current belief state and dynamically select high-impact regions in the graph, it is expected that $\rho_t \geq \rho_{\text{static}}$ in each round. When the graph structure allows the agent to prune irrelevant branches based on early evidence, this inequality becomes strict with non-zero probability. Summing over all rounds, the total information gain of the adaptive strategy satisfies:

$$\mathbb{E}_\pi \left[ \sum_{t=1}^{T} I(A^\star; \mathcal{E}_t \mid Q, \mathcal{H}_{t-1}) \right] = \mathbb{E}_\pi \left[ I(A^\star; \mathcal{H}_T \mid Q) \right] \geq I_{\text{static}},$$ (25)

with strict inequality under the above conditions. From a Bayesian viewpoint, retrieval efficiency can be seen as how much uncertainty is reduced per token. Because the adaptive policy achieves a greater entropy reduction under the same budget, or requires fewer tokens to reach the same posterior certainty, it is strictly more efficient. Moreover, by Fano's inequality,

$$P_e \leq \frac{H(A^\star \mid Q) - I(A^\star; \mathcal{H}_T \mid Q) + 1}{\log |\mathcal{A}|},$$ (26)

we conclude that the lower the conditional entropy, the lower the expected error. Therefore, greater mutual information directly translates into improved answer accuracy.

In conclusion, multi-turn interaction enables the agent to reason over what has already been retrieved, selectively expanding into the most informative parts of the graph, leading to more efficient and accurate question answering. $\qquad\square$

### B.3 PROOF OF PROPOSITION 3

**Proposition 3.** *End-to-end RL bridges the gap between graph-based knowledge and language.*

*Proof.* Let $G_H$ denote the graph-structured knowledge base, and $q$ a given query. The agent (parameterized by $\theta$) interacts over multiple steps, forming a trajectory $\tau = (s_1, a_1, \ldots, s_T, a_T)$ where each $a_t$ is either a graph query or a natural-language output. The policy induces an answer distribution:

$$P_\theta(y \mid q, G_H) = \sum_{\tau:\, \text{answer}(\tau)=y} \pi_\theta(\tau \mid q, G_H).$$ (27)

To align graph usage with answer generation, we define a trajectory-level reward:

$$R(\tau) = r_{\text{fmt}}(\tau) + \mathbb{I}\{r_{\text{fmt}}(\tau) = 1\} \cdot r_{\text{ans}}(y_T, y_q^\star) - 1,$$ (28)

where $r_{\text{fmt}}$ ensures proper structure (e.g., retrieve before answer), and $r_{\text{ans}}$ measures answer quality. Only valid, grounded answers receive positive reward. The expected reward is maximized via policy gradient:

$$\nabla_\theta J(\theta) \propto \mathbb{E}_{\tau \sim \pi_\theta} \left[ \sum_{t=1}^{T} \nabla_\theta \log \pi_\theta(a_t \mid s_t; G_H) \cdot \hat{A}(\tau) \right],$$ (29)

where $\hat{A}(\tau)$ derives from $R(\tau)$. Trajectories that retrieve the right subgraph and generate correct answers are reinforced, linking graph retrieval to linguistic accuracy. As training progresses, the expected log-likelihood of the gold answer increases:

$$\mathcal{L}_\theta = -\log \sum_\tau \pi_\theta(\tau \mid q, G_H) P(y_q^\star \mid \tau),$$ (30)

which lower-bounds the ideal $\log P^\star(y_q^\star \mid q, G_H)$. In the limit, we approach:

$$P_\theta(\cdot \mid q, G_H) \rightarrow P^\star(\cdot \mid q, G_H).$$ (31)

This also manifests as a reduction in conditional entropy:

$$H_\theta(Y \mid Q, G_H) < H(Y \mid Q),$$ (32)

since ungrounded answers are discouraged and graph-consistent ones are promoted. By Fano's inequality, lower entropy implies lower error.

Thus, end-to-end RL not only learns to query the graph but also binds retrieved knowledge to answer generation, effectively bridging the gap between structure and language. $\qquad\square$

# C  DETAILS OF KNOWLEDGE HYPERGRAPH CONSTRUCTION

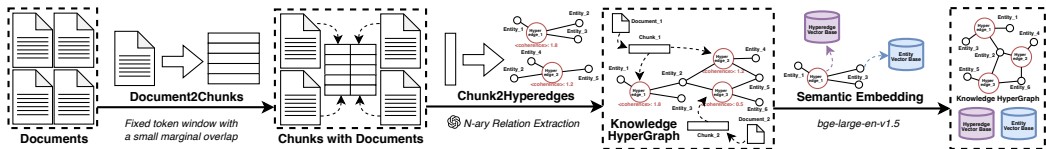

Figure 11: Knowledge HyperGraph Construction in Graph-R1.

As shown in Figure 11, Graph-R1 constructs the Knowledge HyperGraph through a streamlined yet expressive three-stage pipeline.

**First**, we segment raw documents into fixed-size textual units to maintain stable local semantics. Specifically, each document is divided using a 1200-token window with a 50-token overlap, ensuring that adjacent chunks retain sufficient contextual continuity. This windowing strategy preserves semantically meaningful boundaries while preventing the loss of cross-sentence information.

**Second**, each chunk is processed using a GPT-4o-mini–based n-ary relation extraction module. We adopt the carefully designed prompt shown in Appendix A.1, Figure 9, which guides the model to identify sets of entities and the higher-order relations binding them. Each extracted relational fact is represented as a hyperedge, and we assign a coherence score that reflects the internal semantic consistency and reliability of the extracted n-ary relation. This step transforms linear text into a structured hypergraph where nodes represent entities and hyperedges capture rich n-ary facts.

**Third**, to enable semantic retrieval and reasoning over the constructed hypergraph, we generate dense embeddings for both entities and hyperedges using the large-scale embedding model bge-large-en-v1.5. The resulting entity vector base and hyperedge vector base form the foundation of Graph-R1's retrieval module, allowing subsequent reasoning steps.

Compared with the original HyperGraphRAG implementation, we introduce several architectural and engineering optimizations, particularly in parallelization, prompt efficiency, storage layout, and vectorization. These improvements reduce token consumption and computational overhead, yielding a more efficient hypergraph construction pipeline, as evidenced in Table 3 of Section 5.4.

# D  MORE EXPERIMENTAL RESULTS ON O.O.D. SETTINGS

To rigorously assess the out-of-domain (O.O.D.) robustness and cross-domain generalization of Graph-R1, we evaluate the model under three distinct training regimes, each designed to isolate a different aspect of domain transferability. These settings provide a comprehensive picture of how retrieval structures (chunks vs. hypergraphs) behave when training and testing domains differ.

**(i) In-Domain (I.I.D.) Training Setting.** The first setting follows the same configuration as in Table 2. Here, each model is trained exclusively on the training split of its own dataset, meaning the supervision is fully aligned with the target domain. This represents the upper-bound scenario in which rich in-domain training data are available, allowing us to evaluate whether Graph-R1 can surpass Search-R1 even without requiring cross-domain generalization.

**(ii) Single O.O.D. Training Setting.** The second setting corresponds to the cross-dataset training paradigm illustrated in Figures 8. For each target dataset, the model is trained not on its own data, but on the training split of a different dataset. We use the results and compute the average performance. This setting stresses the ability to extract transferable information and tests whether hypergraph-based retrieval yields more robust inductive biases than chunk-based retrieval under dataset shift.

**(iii) Combined-Dataset Training Setting.** The third regime evaluates the model in a mixed-domain learning scenario. We first merge all six datasets into a single training pool and then uniformly downsample 1/6 of the combined data to ensure that the total training data volume remains identical to the I.I.D. setting. The resulting training set contains balanced signals drawn from diverse domains while strictly controlling for data quantity. This design allows us to examine whether Graph-R1 can exploit heterogeneous multi-domain relational structures to improve generalization, while ruling out gains attributable merely to increased training size.

## D.1 RESULTS AND ANALYSIS ON SIX OPEN-DOMIN DATASETS (TABLE 5)

| Method | 2Wiki. | | HotpotQA | | Musique | | NQ | | PopQA | | TriviaQA | | Avg. | |
|---|---|---|---|---|---|---|---|---|---|---|---|---|---|---|
| | EM | F1 | EM | F1 | EM | F1 | EM | F1 | EM | F1 | EM | F1 | EM | F1 |
| *Qwen2.5-3B-Instruct (Results When Trained on Its Own I.I.D. Dataset)* | | | | | | | | | | | | | | |
| Search-R1 (3B) | 31.25 | 38.04 | 38.28 | 43.84 | 3.91 | 7.65 | 24.22 | 37.96 | 33.59 | 38.67 | 40.62 | 47.99 | 28.65 | 35.69 |
| Graph-R1 (3B) | **50.00** | **57.56** | **50.78** | **56.75** | **32.81** | **40.51** | **30.47** | **44.75** | 37.50 | **45.65** | **53.13** | **62.31** | **42.45** | **51.26** |
| *Qwen2.5-3B-Instruct (Average Performance When Trained on a Single O.O.D. Dataset)* | | | | | | | | | | | | | | |
| Search-R1 (3B) | – | 24.30 | – | 29.40 | – | 5.64 | – | 25.46 | – | 26.59 | – | 40.29 | – | 25.28 |
| Graph-R1 (3B) | – | 42.65 | – | 50.20 | – | 32.06 | – | 38.20 | – | 42.63 | – | 58.85 | – | 44.10 |
| *Qwen2.5-3B-Instruct (Results When Trained on Combined Datasets)* | | | | | | | | | | | | | | |
| Search-R1 (3B) | 25.00 | 29.68 | 36.72 | 41.59 | 10.16 | 15.48 | 21.09 | 34.78 | 35.16 | 39.07 | 42.97 | 50.92 | 28.52 | 35.25 |
| Graph-R1 (3B) | 42.97 | 51.07 | 46.88 | 54.97 | 32.03 | 38.29 | 27.34 | 41.00 | **38.28** | 45.32 | 50.00 | 58.49 | 39.58 | 48.19 |

Table 5: Comparison between Search-R1 (3B) and Graph-R1 (3B) across six multi-hop and open-domain datasets under three training regimes. Best results are in **bold** and second in underline.

Table 5 reports the performance of Search-R1 and Graph-R1 under three training regimes (I.I.D., single O.O.D., and combined), covering six multi-hop and open-domain datasets.

First, Graph-R1 achieves consistent and clear improvements over Search-R1 across all datasets and all training settings. This confirms the advantage of hypergraph-structured retrieval, which provides more coherent and relationally grounded evidence than chunk-based retrieval, leading to stronger reasoning and higher EM/F1 scores.

Second, comparing the three regimes, the results follow a stable pattern: I.I.D. training performs best, combined-dataset training ranks second, and single O.O.D. training performs worst. This indicates that dataset-aligned supervision remains the most effective, but exposure to balanced multi-dataset data is still substantially better than training on a completely mismatched dataset.

## D.2 ADDITIONAL O.O.D. RESULTS ON FIVE SPECIALIZED DOMAINS (TABLE 6)

| Method | Medicine | | Agriculture | | CS | | Legal | | Mix | | Avg. | |
|---|---|---|---|---|---|---|---|---|---|---|---|---|
| | EM | F1 | EM | F1 | EM | F1 | EM | F1 | EM | F1 | EM | F1 |
| *GPT-4o-mini* | | | | | | | | | | | | |
| NaiveGeneration | – | 12.89 | – | 12.74 | – | 18.65 | – | 21.64 | – | 16.93 | – | 16.57 |
| StandardRAG | – | 27.90 | – | 27.43 | – | 28.93 | – | 37.34 | – | 43.20 | – | 32.96 |
| GraphRAG | – | 17.60 | – | 21.28 | – | 23.33 | – | 30.11 | – | 19.27 | – | 22.32 |
| LightRAG | – | 12.79 | – | 18.24 | – | 22.72 | – | 31.64 | – | 27.03 | – | 22.48 |
| PathRAG | – | 14.94 | – | 21.30 | – | 26.73 | – | 31.29 | – | 37.07 | – | 26.27 |
| HippoRAG2 | – | 21.34 | – | 12.63 | – | 17.34 | – | 18.53 | – | 21.53 | – | 18.27 |
| HyperGraphRAG | – | 35.35 | – | 33.89 | – | 31.30 | – | 43.81 | – | **48.71** | – | 38.61 |
| *Qwen2.5-3B-Instruct (Zero-Shot Results Without Domain Training)* | | | | | | | | | | | | |
| Search-R1 (3B) | 12.70 | 21.84 | 11.33 | 17.65 | 17.97 | 25.39 | 14.65 | 24.84 | 23.83 | 31.90 | 16.10 | 24.32 |
| Graph-R1 (3B) | **24.22** | **37.18** | **25.98** | **37.39** | **27.54** | **37.45** | **21.48** | 33.66 | **38.48** | **50.60** | **27.54** | **39.26** |

Table 6: Domain-wise results on five specialized datasets. Results of GPT-4o-mini baselines are reported from the HyperGraphRAG paper. Best results are in **bold** and second in underline.

To further assess zero-shot generalization, Table 6 evaluates the model, trained only under the combined setting, on five specialized domain datasets introduced in HyperGraphRAG. For GPT-4o-mini baselines, we report the original results from the HyperGraphRAG paper.

Across Medicine, Agriculture, CS, Legal, and the Mixed domain, Graph-R1 (3B) consistently surpasses Search-R1 by sizeable margins in both EM and F1. Notably, Graph-R1 delivers strong improvements even in knowledge-intensive fields (e.g., Medicine and CS), where structured multi-entity relationships are crucial. These findings suggest that the relational inductive biases encoded by the hypergraph representation transfer effectively across specialized verticals not seen during training.

Collectively, these experiments demonstrate that Graph-R1 offers superior cross-domain robustness, strong transferability to unseen fields, and stable zero-shot performance, significantly outperforming chunk-based RAG baselines across both horizontal (general) and vertical (specialized) domains.

# E  GRAPH-R1 ALGORITHM DETAILS

To illustrate the mechanism of Graph-R1, we present its full workflow in Algorithm 1, comprising three key phases: **(1) Hypergraph Construction.** An LLM-based extractor $\pi_{\text{ext}}$ extracts $n$-ary relational facts from the corpus $K$ to build a semantic hypergraph $\mathcal{G}_H = (V, E_H, \phi)$, where all elements are encoded via $\text{Enc}(\cdot)$. **(2) Multi-turn Agentic Reasoning.** Given query $q$, the agent performs reflection, intent selection, and action generation over $T$ steps under policy $\pi_\theta$. Queries trigger dual-path hypergraph retrieval; answers terminate reasoning. **(3) End-to-end RL Optimization.** The policy is optimized using GRPO, guided by format and answer rewards. The objective $\mathcal{J}_{\text{GRPO}}$ is computed from sampled trajectories $\{\tau_i\}$ with clipped advantage-weighted updates.

---

**Algorithm 1** Graph-R1: Agentic GraphRAG via End-to-end RL

---

**Require:** Query $q$, knowledge corpus $K = \{d_1, \ldots, d_N\}$, policy $\pi_\theta$, reward function $R(\tau)$
**Ensure:** Final answer $y_q$

1: **// 1: Knowledge Hypergraph Construction**
2: Initialize hypergraph $\mathcal{G}_H = (V, E_H, \phi)$
3: **for** each document $d \in K$ **do**
4:     Extract relational facts: $\{(h_i, \mathcal{V}_{h_i})\} \sim \pi_{\text{ext}}(d)$
5:     **for** each $(h_i, \mathcal{V}_{h_i})$ **do**
6:         $E_H \leftarrow E_H \cup \{h_i\}$,   $V \leftarrow V \cup \mathcal{V}_{h_i}$
7:         $\phi(h_i) \leftarrow \text{Enc}(h_i)$,   $\phi(v) \leftarrow \text{Enc}(v)$ for $v \in \mathcal{V}_{h_i}$
8:     **end for**
9: **end for**

10: **// 2: Multi-turn Graph Reasoning**
11: Initialize state $s_1 \leftarrow q$, trajectory $\tau \leftarrow \emptyset$
12: **for** $t = 1$ to $T$ **do**
13:     Generate reasoning plan: $\mathbf{a}_t^{\text{think}} \sim \pi_\theta(\cdot \mid s_t)$
14:     Choose intent: $\alpha_t \sim \pi_\theta(\cdot \mid \mathbf{a}_t^{\text{think}}, s_t)$
15:     **if** $\alpha_t = (\text{answer})$ **then**
16:         Output answer: $\mathbf{a}_t^{\text{ans}} \sim \pi_\theta(\cdot \mid \mathbf{a}_t^{\text{think}}, s_t)$
17:         $\tau \leftarrow \tau \cup \{(s_t, \mathbf{a}_t^{\text{query}}, \mathbf{a}_t^{\text{ans}})\}$;   **return** $y_q = \mathbf{a}_t^{\text{ans}}$
18:     **else if** $\alpha_t = (\text{query, retrieve})$ **then**
19:         Generate query: $\mathbf{a}_t^{\text{query}} \sim \pi_\theta(\cdot \mid \mathbf{a}_t^{\text{think}}, s_t)$
20:         Entity retrieval: $\mathcal{R}_V = \text{argmax}_{v \in V}^{k_V} \text{sim}(\phi(v), \phi(V_{\mathbf{a}_t^{\text{query}}}))$
21:         Hyperedge retrieval: $\mathcal{R}_H = \text{argmax}_{h \in E_H}^{k_H} \text{sim}(\phi(h), \phi(\mathbf{a}_t^{\text{query}}))$
22:         Rank fusion: $\mathbf{a}_t^{\text{ret}} = \text{Top-}k\left(\mathcal{F}_V^* \cup \mathcal{F}_H^*, \text{ Score}(f) = \frac{1}{r_V(f)} + \frac{1}{r_H(f)}\right)$
23:         Update state $s_{t+1} \leftarrow s_t \cup \{(s_t, \mathbf{a}_t^{\text{think}}, \mathbf{a}_t^{\text{query}}, \mathbf{a}_t^{\text{ret}})\}$
24:         $\tau \leftarrow \tau \cup \{(s_t, \mathbf{a}_t^{\text{think}}, \mathbf{a}_t^{\text{query}}, \mathbf{a}_t^{\text{ret}})\}$
25:     **end if**
26: **end for**

27: **// 3: End-to-end Policy Optimization (GRPO)**
28: Sample $N$ trajectories $\{\tau_i\} \sim \pi_{\theta_{\text{old}}}$
29: **for** each $\tau_i$ **do**
30:     Compute reward: $R(\tau_i) = -1 + R_{\text{format}}(\tau_i) + \mathbb{I}\{R_{\text{format}} = 1\} \cdot R_{\text{answer}}(y_T, y_q^\star)$
31:     Compute advantage: $\hat{A}(\tau_i) = \frac{R(\tau_i) - \text{mean}(\{R(\tau_j)\})}{\text{std}(\{R(\tau_j)\})}$
32: **end for**
33: Update policy via GRPO: $\mathcal{J}_{\text{GRPO}} \sim \sum_{i=1}^{N} \sum_{t=1}^{|\tau_i|} \min\left(\rho_\theta(a_t^{(i)})\hat{A}(\tau_i), \text{clip}(\rho_\theta(a_t^{(i)}), 1 \pm \epsilon)\hat{A}(\tau_i)\right)$

34: where $\rho_\theta(a_t^{(i)}) = \frac{\pi_\theta(a_t^{(i)} \mid s_{t-1}^{(i)})}{\pi_{\theta_{\text{old}}}(a_t^{(i)} \mid s_{t-1}^{(i)})}$

---

*Complexity Analysis.* Graph-R1 involves three computational components corresponding to phases. First, hypergraph construction scales with the total token count $T_K$ of the knowledge corpus and the number of extracted relational facts $F$, yielding complexity $O(T_K) + O(F)$. Second, during multi-turn reasoning, the agent performs $T$ steps of action sampling and dual-path retrieval. At each step, similarity computations over $|V|$ nodes and $|E_H|$ hyperedges with embedding dimension $d$ yield $O((|V| + |E_H|)d)$ per step. Third, for policy optimization, GRPO processes $N$ sampled trajectories of max length $T$, with gradient updates costing $O(NTd)$. Each component is computationally tractable and benefits from parallelization and localized retrieval over compact hypergraph subsets.

## F    DATASET DETAILS

We conduct experiments on six widely-used RAG benchmarks selected from the FlashRAG toolkit (Jin et al., 2025b), covering both single-hop and multi-hop question answering tasks:

- **2WikiMultiHopQA (2Wiki.)** (Ho et al., 2020): A multi-hop dataset requiring reasoning across two Wikipedia documents.
- **HotpotQA** (Yang et al., 2018): A challenging multi-hop QA dataset with sentence-level supporting facts and diverse question types.
- **Musique** (Trivedi et al., 2022): Multi-hop questions needing chains of inference, often involving three or more reasoning steps.
- **Natural Questions (NQ)** (Kwiatkowski et al., 2019): A large-scale single-hop QA dataset grounded in real Google search questions with Wikipedia passages.
- **PopQA** (Mallen et al., 2023): An open-domain QA dataset focused on popular culture questions sourced from Wikipedia.
- **TriviaQA** (Joshi et al., 2017): A large-scale dataset containing trivia-style questions with distantly supervised evidence documents.

To ensure consistency across datasets and maintain manageable training and evaluation workloads, we uniformly sample 5,120 instances per dataset for training and 128 instances for testing.

## G    BASELINE DETAILS

Our experiments compare Graph-R1 with two groups of baselines using different backbone LLMs:

### G.1    BASELINES WITH GPT-4O-MINI

- **NaiveGeneration (GPT-4o-mini)**: Zero-shot generation using GPT-4o-mini without retrieval, evaluating base model capacity.
- **StandardRAG (GPT-4o-mini)** (Lewis et al., 2020): Chunk-based RAG using GPT-4o-mini as the generator with retrieval over text chunks.
- **GraphRAG** (Edge et al., 2025): Graph-structured retrieval baseline that constructs entity graphs and performs one-shot retrieval with GPT-4o-mini for answer generation.
- **LightRAG** (Guo et al., 2025): A lightweight GraphRAG variant that builds compact graphs for more efficient retrieval and GPT-4o-mini generation.
- **PathRAG** (Chen et al., 2025): Retrieval via path-based pruning on entity graphs, followed by GPT-4o-mini answer synthesis.
- **HippoRAG2** (Gutiérrez et al., 2025): Hierarchical path planner over knowledge graphs to improve retrieval efficiency, with GPT-4o-mini used for generation.
- **HyperGraphRAG** (Luo et al., 2025a): Constructs n-ary relational hypergraphs to support a single retrieval step, and uses GPT-4o-mini for answer writing.

### G.2    BASELINES WITH QWEN2.5 (1.5B, 3B, 7B)

- **NaiveGeneration**: Direct generation by Qwen2.5 given the question prompt, without any retrieval, serving as a lower bound baseline.
- **StandardRAG** (Lewis et al., 2020): Classic chunk-based retrieval-augmented generation pipeline with semantic retriever and Qwen2.5 decoder.
- **SFT** (Zheng et al., 2024): Supervised fine-tuning of Qwen2.5 on QA pairs, without multi-turn reasoning or reinforcement optimization.
- **R1** (Shao et al., 2024): A GRPO-trained policy that generates final answers directly from question prompts without retrieval, optimized only on answer quality.
- **Search-R1** (Jin et al., 2025a): A multi-turn chunk-based retrieval method trained with GRPO, capable of iterative query refinement and retrieval under a unified policy.
- **R1-Searcher** (Song et al., 2025): A two-stage GRPO-based method with chunk-based retrieval: first using only format-level rewards to produce structured traces, then adding answer rewards.

## H EVALUATION DETAILS

Inspired by Luo et al. (2025a); Jin et al. (2025a), we evaluate model performance using four metrics:

**(i) Exact Match (EM).** EM measures whether the predicted answer exactly matches the ground truth. Let $\text{norm}(\cdot)$ denote the normalization function:

$$\text{EM} = \frac{1}{N} \sum_{i=1}^{N} \mathbb{I}\left\{\text{norm}(y_i) = \text{norm}(y_i^\star)\right\}. \quad (33)$$

**(ii) F1 Score.** The F1 score measures the token-level overlap between the predicted answer $y_i$ and the ground-truth answer $y^\star$ using the harmonic mean of precision and recall:

$$\text{F1} = \frac{1}{N} \sum_{i=1}^{N} \frac{2 \cdot |\text{tokens}(y_i) \cap \text{tokens}(y_i^\star)|}{|\text{tokens}(y_i)| + |\text{tokens}(y_i^\star)|}. \quad (34)$$

**(iii) Retrieval Similarity (R-S).** R-S assesses semantic similarity between retrieved $k_{\text{retr}}^{(i)}$ and gold knowledge $k_{\text{retr}}^{(i)}$. Let $\text{Enc}(\cdot)$ be the semantic embedding function:

$$\text{R-S} = \frac{1}{N} \sum_{i=1}^{N} \cos\left(\text{Enc}(k_{\text{retr}}^{(i)}), \text{Enc}(k_{\text{gold}}^{(i)})\right). \quad (35)$$

**(iv) Generation Evaluation (G-E).** G-E reflects generation quality. Let $s_{i,d}$ be GPT-4o-mini scores across 7 criteria (Figure 12):

$$\text{G-E} = \frac{1}{N} \sum_{i=1}^{N} \left(\frac{1}{7} \sum_{d=1}^{7} s_{i,d}\right). \quad (36)$$

Figure 12: Seven Dimensions for Generation Evaluation.

## I IMPLEMENTATION DETAILS

As shown in Table 7, we summarize the detailed hyperparameter configurations used throughout our experiments, including model backbone, input limits, training configuration, and retrieval setup.

| Method | Backbone | Batch Size | Max Length | Top-K | Algo | Epochs |
|---|---|---|---|---|---|---|
| NaiveGeneration | Qwen2.5 / GPT-4o-mini | – | $\infty$ | N/A | – | – |
| StandardRAG | Qwen2.5 / GPT-4o-mini | – | $\infty$ | 5 Chunks | – | – |
| GraphRAG | GPT-4o-mini | – | $\infty$ | 60 | – | – |
| LightRAG | GPT-4o-mini | – | $\infty$ | 60 | – | – |
| PathRAG | GPT-4o-mini | – | $\infty$ | 60 | – | – |
| HippoRAG2 | GPT-4o-mini | – | $\infty$ | 60 | – | – |
| HyperGraphRAG | GPT-4o-mini | – | $\infty$ | 60 | – | – |
| SFT | Qwen2.5 (1.5B, 3B, 7B) | 16 | 4096 | N/A | LoRA | 3 |
| R1 | Qwen2.5 (1.5B, 3B, 7B) | 128 | 4096 | N/A | GRPO | 1 |
| Search-R1 | Qwen2.5 (1.5B, 3B, 7B) | 128 | 4096 | 5 Chunks / Turn | GRPO | 1 |
| R1-Searcher | Qwen2.5 (1.5B, 3B, 7B) | 128 | 4096 | 5 Chunks / Turn | GRPO | 1 |
| Graph-R1 (ours) | Qwen2.5 (1.5B, 3B, 7B) | 128 | 4096 | 5 / Turn | GRPO | 1 |

Table 7: Hyperparameter settings for baselines and Graph-R1.

## J LIMITATIONS AND FUTURE WORK

While Graph-R1 achieves strong performance, several limitations remain. **First**, the cost of hypergraph construction, especially relation extraction and encoding, remains non-trivial. Future work may explore more efficient methods for zero-cost extraction. **Second**, current retrieval lacks structural reasoning. Integrating GNNs or trainable message-passing could improve both accuracy and scalability. **Third**, Graph-R1 currently supports only textual knowledge; extending it to multi-modal inputs is a promising direction. **Finally**, we aim to further apply Graph-R1 in knowledge-intensive domains such as healthcare, law, and finance, where robust and interpretable reasoning is essential.

