# OpenReview forum: "Graph-R1: Towards Agentic GraphRAG Framework via End-to-end Reinforcement Learning"
_ICLR.cc/2026/Conference — Submitted to ICLR 2026_

### Official Review · Reviewer_g7wu · 2025-10-26

**Soundness:** 4
**Presentation:** 2
**Contribution:** 3
**Rating:** 6
**Confidence:** 4

**Summary:**

The paper presents Graph-R1, an optimized agent for retrieval-augmented generation using GraphRAG. Graph-R1 first constructs a HyperGraph that serves as its backbone retrieval system, enabling structured knowledge retrieval. The agent then employs a multi-turn approach to answer queries by iteratively refining search queries and hypergraph retrieval operations (entity/hyperedge retrieval), allowing for progressive improvement in response quality. Graph-R1 is trained end-to-end using GRPO reinforcement learning, with rewards designed to encourage both correct answer generation and proper response formatting. Experimental evaluation on six datasets demonstrates Graph-R1's effectiveness, significantly outperforming competing GraphRAG approaches and search agents across metrics.

**Strengths:**

- S1) Graph-R1 demonstrates strong performance, outperforming existing GraphRAG approaches and search agents across multiple benchmarks.

- S2) Ablation studies show that combining enhanced knowledge representations (GraphRAG) with reinforcement learning yields greater performance gains than traditional chunk-based retrieval methods.

- S3) The paper provides comprehensive experimental analysis across multiple dimensions, such as performance, efficiency, response length, and turn count, demonstrating Graph-R1's advantages.

**Weaknesses:**

- W1) Graph-R1 appears to rely on existing approaches, Search-R1 (Jin et al., 2025) and HyperGraphRAG (Luo et al., 2025), i.e., on Search-R1's reinforcement learning framework and HyperGraphRAG's retrieval operations and graph construction. In addition, HyperGraphRAG seems to outperform Graph-R1 on some benchmarks when using GPT-4o-mini (Table 2), while Graph-R1 requires >15 steps to surpass HyperGraphRAG's performance (Figure 5b).

- W2) Graph-R1's reliance on entity-based retrieval (Section 2.2) may limit its generalizability to queries lacking explicit entities, such as "Summarize the trends in the last 5 financial reports."

**Questions:**

- Q1) Do Graph-R1 and Search-R1 use identical training data, or are you reporting results from the released Search-R1 version? Additionally, what is the number of training examples per method?

---

> ### Author Response · Authors · 2025-11-21
> **Response to Reviewer g7wu (Part 1/2)**
>
> Thank you very much for your time and effort in reviewing our paper. We sincerely appreciate your feedback. Below, we respectfully provide our detailed responses to address your concerns.
>
> ---
> **W1a: "Graph-R1 appears to rely on existing approaches, Search-R1 (Jin et al., 2025) and HyperGraphRAG (Luo et al., 2025)."**
>
> Thank you for the thoughtful feedback. To clarify our contributions, we summarize them from the following **five perspectives**:
>
> **a. Motivation Level:**
>   - Prior GraphRAG systems rely on passive, one-shot retrieval, and **LLMs struggle to interpret retrieved graph structures**, often performing worse than chunk-based RAG.
>   - We identify GraphRAG’s **structural bottlenecks** (**high-cost** graph construction, **fixed** retrieval without reasoning, and a **gap** between graph representations and LLM reasoning), and want to **construct the first learnable, multi-turn agent that actively explores graph structures**, enabling reasoning over structured knowledge hypergraphs.
>
> **b. Technical Level:**
> - GraphRAG comprises **three core modules**—**construction**, **retrieval**, and **generation**. Addressing **key challenges in each part**, we introduce innovations that build a **learnable “data–knowledge dual-driven” GraphRAG system**.
> - For **Construction (Knowledge Construction and Agent Initialization)**, we propose a **lightweight n-ary hypergraph construction pipeline** and, for the first time, **formulate the hypergraph as an interactive agent environment** for structural interaction.
> - For **Retrieval (Knowledge Reasoning via Multi-Turn Graph Interaction)**, we develop a **dual-path retrieval mechanism fused via Reciprocal Rank Fusion (RRF)**, combining entity-based and semantic hyperedge-based signals to produce **stable, high-quality subgraph evidence**.
> - For **Generation (Outcome-Directed End-to-End Reinforcement Learning)**, we design a **specialized, outcome-aligned reward scheme** that couples **format-level structural validity** with **answer-level correctness**, ensuring reliable multi-step reasoning.
> - Together, these innovations yield the **first unified, interactive, and optimizable GraphRAG system**, with the effectiveness of each module validated in **Sections 5.4, 5.5, and 5.6**.
>
> **c. Theoretical Level:**
> - We show that RL not only improves answer quality but also refines retrieval trajectories, significantly **narrowing the gap** between **graph structures** and **LLM generation**.
> - **Proof 1: Graph-structured knowledge boosts agent accuracy by richer representation.**
> We support this by modeling the hypergraph as a more informative state space and observing consistent accuracy gains over text-based baselines.
> - **Proof 2: Multi-turn interaction with the graph environment improves retrieval efficiency.**
> We justify this through a state-update view of iterative refinement and empirically by reduced retrieval volume with higher answer quality.
> - **Proof 3: End-to-end RL bridges the gap between graph-based knowledge and language.**
> We validate this through a coupled optimization formulation and improved performance over non-learnable or decoupled pipelines.
> - To the best of our knowledge, this is the first work to **theoretically analyze** and **empirically validate** RL-driven reasoning over knowledge hypergraphs, which has not been explored in previous GraphRAG, Search-R1, or RL-RAG research.
>
> **d. Code Level:**
> - Graph-R1 is **the first open-source end-to-end trainable GraphRAG+RL framework**, supporting graph construction, retrieval, multi-turn agent rollouts, and GRPO training.
> - Unlike prompt-based GraphRAG systems, our framework enables **easily training an agent** that **actively operates within a graph**, providing substantial community value beyond a single algorithm.
>
> **e. Promising Level:**
> - Where current GraphRAG **relies on prompt engineering and long-context inference**, Graph-R1 shows that models can learn to use structured knowledge via RL.
> - This **neuro-symbolic direction** transforms graph knowledge from a passive context source into an active reasoning enhancer, opening a **scalable and learnable** pathway for future GraphRAG systems.
>
> In summary, Graph-R1 introduces **the first end-to-end learnable agentic GraphRAG framework** that actively interacts with hypergraph-structured knowledge, enabling the model to autonomously explore, select, and reason over structured evidence. By unifying hypergraph construction, multi-turn retrieval, and RL-driven reasoning within a single trainable system, our work demonstrates a scalable pathway for models to learn to use structured knowledge rather than rely on handcrafted prompts or fixed retrieval. We believe this establishes a solid foundation for future research on **learnable, neuro-symbolic GraphRAG systems**, and we hope this clarification helps highlight the significance of our contributions.
>
> ---

---

> ### Author Response · Authors · 2025-11-21
> **Response to Reviewer g7wu (Part 2/2)**
>
> ---
> **W1b: "In addition, HyperGraphRAG seems to outperform Graph-R1 on some benchmarks when using GPT-4o-mini (Table 2)."**
>
> - We thank the reviewer for the observation. We would like to clarify the evaluation metrics in **Table 2**:
> 	- F1 measures token-level answer overlap.
> 	- EM measures exact string match.
> 	- R-S measures retrieval similarity using a **pretrained language model** to score semantic alignment between retrieved evidence and gold evidence.
> 	- G-E uses **GPT-4o-mini** as an **LLM-as-a-judge**, evaluating explanation quality and grounding.
>
> - Regarding the reviewer’s concern:
>   - G-E relies on GPT-4o-mini as the judge. This inherently favors methods whose reasoning and style are closer to GPT-based APIs, and is **not fully fair when comparing GPT-4o-mini with Qwen 1.5B/3B/7B models**. As a result, G-E is best compared **within the same model family / parameter scale**, not across heterogeneous model types.
>   - R-S depends on a pretrained encoder whose embedding space is **naturally more preferred** with more powerful GPT-based models. Therefore, comparing GPT-4o-mini–based approaches with Qwen-based models on R-S is not an entirely symmetric setting.
>
> - In contrast, F1 and EM remain the most objective indicators for cross-model comparison, and Graph-R1 **consistently shows strong improvements** on these metrics.
>
> ---
> **W1c: "While Graph-R1 requires >15 steps to surpass HyperGraphRAG's performance (Figure 5b)."**
>
> - We believe this is a misunderstanding. **Figure 5(b)** does not show that Graph-R1 **“requires >15 steps to surpass HyperGraphRAG.”** The x-axis denotes RL **training iterations**, **not inference-time reasoning steps**. It simply reflects the learning curve during optimization, not the number of steps needed to answer a question. Thus, the figure illustrates training efficiency, not inference complexity.
>
> - As also discussed in [1], **reinforcement learning** allows a model to **iteratively self-improve**, and its performance can gradually **surpass** that of larger pretrained models as the agent learns better policies through experience.
>
> >**Reference:**
> [1] Guo, Daya, et al. "Deepseek-r1 incentivizes reasoning in llms through reinforcement learning." Nature 645.8081 (2025): 633-638.
>
> - Moreover, as shown in **Figure 5(b)**, Graph-R1 actually achieves the **fastest RL training efficiency** among all compared methods, reaching higher performance with fewer optimization iterations.
>
> ---
> **W2: "Graph-R1's reliance on entity-based retrieval (Section 2.2) may limit its generalizability to queries lacking explicit entities, such as "Summarize the trends in the last 5 financial reports.""**
>
> - We appreciate the reviewer’s concern. However, Graph-R1 is not limited to entity-based retrieval. As described in **Section 4.2**, our framework retrieves **both entities and hyperedges**.
> - **Each hyperedge** encodes a full n-ary fact expressed in **natural language**, and our agent retrieves these hyperedges directly through **natural-language queries**, not only through explicit entity mentions.
>
> - Moreover, Graph-R1 employs a **rank-fusion mechanism** that combines (i) **entity-based** signals when available, and (ii) **hyperedge-level** semantic matching when entities are absent or ambiguous. This integrated retrieval process enables the agent to generalize to broad, **non-entity-centric queries** while preserving information completeness via hyperedges.
>
> ---
> **Q1: "Do Graph-R1 and Search-R1 use identical training data, or are you reporting results from the released Search-R1 version? Additionally, what is the number of training examples per method?"**
>
> - Graph-R1 and Search-R1 are **trained on exactly the same training data**. We do not use the released Search-R1 model; instead, we retrained Search-R1 from scratch on our unified dataset to ensure a fully fair comparison.
> - As described in **Appendix D (Dataset Details)**, we uniformly sampled **5,120 training examples** per dataset and **128 test examples** per dataset. Both Graph-R1 and Search-R1 use this identical 5,120-example training set for each dataset, and the overall training corpus size is the same across all methods.
> - Thus, all compared models—Graph-R1, Search-R1, and baseline variants—are **trained under identical data volume** to ensure fairness.
>
> ---
> At last, we sincerely appreciate your valuable feedback. We have carefully considered all your suggestions and substantially improved and updated the paper accordingly. If our responses and revisions address your concerns, we would be deeply grateful if you could kindly consider raising the score further to support our work. Thank you very much!

---

> > ### Comment · Reviewer_g7wu · 2025-11-26
> >
> > Thank you for the discussion and the effort to address my questions, such as clarifying W1b, W1c, and Q1.
> >
> > However, I did not quite follow how Graph-R1 differs from the combination of Search-R1 with HyperGraphRAG (response to W1a). For example, from the "Technical Details" perspective, points 1 (Lightweight n-ary hypergraph construction) and 3 (Dual-path hyperedge retrieval design with RRF) are already similar to HyperGraphRAG, while point 4 (Outcome-aligned structural reward design) is typical in RL approaches such as Search-R1. Point 2 (First agent–hypergraph interaction architecture) suggests that the technical novelty lies in combining HyperGraphRAG with RL in a Search-R1 style.
> >
> > Regarding W2, could you explain how hyper-edge retrieval helps to tackle more type of queries, such as summarization ones like "Summarize the trends in the last 5 financial reports." ?
> >
> > Overall, I would maintain my positive score.

---

> ### Author Response · Authors · 2025-11-26
>
> Thank you again for the thoughtful follow-up. To clarify W1a, in Graph-R1 these designs are not standalone components but are placed within a single learnable decision loop, where they interact and depend on one another. This unified setup enables capabilities that prior methods could not exhibit—learnable multi-turn graph reasoning, adaptive retrieval policy optimization, and structure-driven performance gains. **If these mechanisms are separated and put back into their original systems, they do not naturally produce such behaviors.** Moreover, our contribution goes beyond architecture: a substantial part of the work lies in the **theoretical grounding** and large-scale **empirical validation** of this unified paradigm. We are the first to verify, from both theoretical and experimental perspectives, three core propositions: (Proof 1) **graph-structured knowledge boosts agent accuracy by richer representation**, (Proof 2) **multi-turn interaction with the graph environment improves retrieval efficiency**, and (Proof 3) **end-to-end RL bridges the gap between graph-based knowledge and language**. These results support the framework as a whole, to further represent genuinely valuable evidence for the significance of this paradigm.
>
> Regarding W2, our point is that hyperedge retrieval can alleviate the limitations of purely entity-based retrieval by matching **natural-language** queries to **natural-language-based n-ary relational facts**, rather than **relying solely on explicit entity mentions**. However, for queries like “Summarize the trends in the last 5 financial reports,” we acknowledge that this type of **time-dependent summarization** is currently challenging—not only for Graph-R1, but **for virtually all existing GraphRAG methods**.
>
> That said, we believe our framework provides a practical path forward. Such queries could be supported by **integrating more external tools** (e.g., **semantic parsing** or **Text2SQL**) to locate the relevant report range, and then using Graph-R1’s RL loop to learn a workflow-style policy (e.g., **identify time span → retrieve corresponding records → extract facts → summarize**). Because each step can still **combine hypergraph-based evidence to improve retrieval efficiency**, we expect this type of extension to be feasible within our framework, and we view it as a promising direction for future development.
>
> At last, we sincerely appreciate the reviewer’s time, thoughtful insights, and constructive feedback again.

---

### Official Review · Reviewer_ZyZx · 2025-10-29

**Soundness:** 2
**Presentation:** 3
**Contribution:** 2
**Rating:** 4
**Confidence:** 3

**Summary:**

This work applied RL with verified outcome reward to enhance the GraphRAG framework. Specifically, it construct a knowledge hypergraph from a given knowledge source and then let an LLM to perform multi-turn graph retrieval on the constructed hypergraph followed up by reasoning / thinking. These process is improved via GRPO with outcome-directed reward function.

**Strengths:**

- The paper is well writing.

- The paper did detailed empirical experiments across multiple datasets and also provide insightful case studies.

**Weaknesses:**

- The research novelty is quite limited. This work applies GRPO to GraphRAG scenario. The most techniques (GRPO, GraphRAG etc.) utilized in this work is already extensively studied. Basically this paper only proves that GRPO can work in the GraphRAG scenario.

- The proposed OOD setting is not enough to evaluate the generalization ability of the finetuned LLMs. Many datasets such as 2wiki and hotpotqa are all wiki-based datasets. The paper need to evaluate on other domain of data to show the generalization ability.

**Questions:**

- Some claims are too strong. For example, the paper claims that existing GraphRAG all aim to gather sufficient knowledge in a single fixed retrieval. However, there are already some GraphRAG work such as HybGRAG [1] that leverages multi-retrieval strategy. The paper need to do a more comprehensive literature review.


[1] HybGRAG: Hybrid Retrieval-Augmented Generation on Textual and Relational Knowledge Bases](https://aclanthology.org/2025.acl-long.43/) (Lee et al., ACL 2025)

---

> ### Author Response · Authors · 2025-11-21
> **Response to Reviewer ZyZx (Part 1/3)**
>
> Thank you very much for your time and effort in reviewing our paper. We sincerely appreciate your feedback. Below, we respectfully provide our detailed responses to address your concerns.
>
> ---
> **Q1: "The research novelty is quite limited. This work applies GRPO to GraphRAG scenario. Basically this paper only proves that GRPO can work in the GraphRAG scenario."**
>
> Thank you for the thoughtful feedback. To clarify our contributions, we summarize them from the following **five perspectives**:
>
> **a. Motivation Level:**
>   - Prior GraphRAG systems rely on passive, one-shot retrieval, and **LLMs struggle to interpret retrieved graph structures**, often performing worse than chunk-based RAG.
>   - We identify GraphRAG’s **structural bottlenecks** (**high-cost** graph construction, **fixed** retrieval without reasoning, and a **gap** between graph representations and LLM reasoning), and want to **construct the first learnable, multi-turn agent that actively explores graph structures**, enabling reasoning over structured knowledge hypergraphs.
>
> **b. Technical Level:**
> - GraphRAG comprises **three core modules**—**construction**, **retrieval**, and **generation**. Addressing **key challenges in each part**, we introduce innovations that build a **learnable “data–knowledge dual-driven” GraphRAG system**.
> - For **Construction (Knowledge Construction and Agent Initialization)**, we propose a **lightweight n-ary hypergraph construction pipeline** and, for the first time, **formulate the hypergraph as an interactive agent environment** for structural interaction.
> - For **Retrieval (Knowledge Reasoning via Multi-Turn Graph Interaction)**, we develop a **dual-path retrieval mechanism fused via Reciprocal Rank Fusion (RRF)**, combining entity-based and semantic hyperedge-based signals to produce **stable, high-quality subgraph evidence**.
> - For **Generation (Outcome-Directed End-to-End Reinforcement Learning)**, we design a **specialized, outcome-aligned reward scheme** that couples **format-level structural validity** with **answer-level correctness**, ensuring reliable multi-step reasoning.
> - Together, these innovations yield the **first unified, interactive, and optimizable GraphRAG system**, with the effectiveness of each module validated in **Sections 5.4, 5.5, and 5.6**.
>
> **c. Theoretical Level:**
> - We show that RL not only improves answer quality but also refines retrieval trajectories, significantly **narrowing the gap** between **graph structures** and **LLM generation**.
> - **Proof 1: Graph-structured knowledge boosts agent accuracy by richer representation.**
> We support this by modeling the hypergraph as a more informative state space and observing consistent accuracy gains over text-based baselines.
> - **Proof 2: Multi-turn interaction with the graph environment improves retrieval efficiency.**
> We justify this through a state-update view of iterative refinement and empirically by reduced retrieval volume with higher answer quality.
> - **Proof 3: End-to-end RL bridges the gap between graph-based knowledge and language.**
> We validate this through a coupled optimization formulation and improved performance over non-learnable or decoupled pipelines.
> - To the best of our knowledge, this is the first work to **theoretically analyze** and **empirically validate** RL-driven reasoning over knowledge hypergraphs, which has not been explored in previous GraphRAG, Search-R1, or RL-RAG research.
>
> **d. Code Level:**
> - Graph-R1 is **the first open-source end-to-end trainable GraphRAG+RL framework**, supporting graph construction, retrieval, multi-turn agent rollouts, and GRPO training.
> - Unlike prompt-based GraphRAG systems, our framework enables **easily training an agent** that **actively operates within a graph**, providing substantial community value beyond a single algorithm.
>
> **e. Promising Level:**
> - Where current GraphRAG **relies on prompt engineering and long-context inference**, Graph-R1 shows that models can learn to use structured knowledge via RL.
> - This **neuro-symbolic direction** transforms graph knowledge from a passive context source into an active reasoning enhancer, opening a **scalable and learnable** pathway for future GraphRAG systems.
>
> In summary, Graph-R1 introduces **the first end-to-end learnable agentic GraphRAG framework** that actively interacts with hypergraph-structured knowledge, enabling the model to autonomously explore, select, and reason over structured evidence. By unifying hypergraph construction, multi-turn retrieval, and RL-driven reasoning within a single trainable system, our work demonstrates a scalable pathway for models to learn to use structured knowledge rather than rely on handcrafted prompts or fixed retrieval. We believe this establishes a solid foundation for future research on **learnable, neuro-symbolic GraphRAG systems**, and we hope this clarification helps highlight the significance of our contributions.
>
> ---

---

> ### Author Response · Authors · 2025-11-21
> **Response to Reviewer ZyZx (Part 2/3)**
>
> ---
> **Q2: "The proposed OOD setting is not enough to evaluate the generalization ability of the finetuned LLMs. Many datasets such as 2wiki and hotpotqa are all wiki-based datasets. The paper need to evaluate on other domain of data to show the generalization ability."**
>
> - Thank you for raising this important point. We fully agree that evaluations limited to Wikipedia-style datasets may not fully reflect cross-domain generalization. To address this concern, we have substantially **expanded our O.O.D. evaluation in the revised version** by adding **Appendix D.2** and **Table 6**, which evaluates Graph-R1 on **five specialized, non-Wikipedia domains** introduced in the HyperGraphRAG benchmark: **Medicine, Agriculture, Computer Science, Legal, Mixed-domain**. All these datasets are entirely outside the Wikipedia domain, providing a rigorous test of cross-domain transfer.
>
> - For clarity, we present **Table 6** below:
>
> | Method         | Medicine EM/F1 | Agriculture EM/F1 | CS EM/F1 | Legal EM/F1 | Mix EM/F1 | Avg. EM/F1 |
> |----------------|----------------|--------------------|----------|--------------|------------|-------------|
> | **GPT-4o-mini** baselines (from HyperGraphRAG paper) |||||||
> | NaiveGeneration | — / 12.89 | — / 12.74 | — / 18.65 | — / 21.64 | — / 16.93 | — / 16.57 |
> | StandardRAG | — / 27.90 | — / 27.43 | — / 28.93 | — / 37.34 | — / 43.20 | — / 32.96 |
> | GraphRAG | — / 17.60 | — / 21.28 | — / 23.33 | — / 30.11 | — / 19.27 | — / 22.32 |
> | LightRAG | — / 12.79 | — / 18.24 | — / 22.72 | — / 31.64 | — / 27.03 | — / 22.48 |
> | PathRAG | — / 14.94 | — / 21.30 | — / 26.73 | — / 31.29 | — / 37.07 | — / 26.27 |
> | HippoRAG2 | — / 21.34 | — / 12.63 | — / 17.34 | — / 18.53 | — / 21.53 | — / 18.27 |
> | HyperGraphRAG | — / 35.35 | — / 33.89 | — / 31.30 | — / **43.81** | — / 48.71 | — / 38.61 |
> | **Qwen2.5-3B-Instruct** (Zero-Shot, Combined-Trained) |||||||
> | Search-R1 (3B) | 12.70 / 21.84 | 11.33 / 17.65 | 17.97 / 25.39 | 14.65 / 24.84 | 23.83 / 31.90 | 16.10 / 24.32 |
> | **Graph-R1 (3B)** | **24.22 / 37.18** | **25.98 / 37.39** | **27.54 / 37.45** | **21.48** / 33.66 | **38.48 / 50.60** | **27.54 / 39.26** |
>
> - **Table 6** evaluates the model, trained only under the combined setting, on five specialized domain datasets introduced in HyperGraphRAG. For GPT-4o-mini baselines, we report the original results from the HyperGraphRAG paper.
>
> - Across Medicine, Agriculture, CS, Legal, and the Mixed domain, **Graph-R1 (3B) consistently surpasses Search-R1** by sizeable margins in both EM and F1. Notably, Graph-R1 delivers strong improvements even in knowledge-intensive fields (e.g., Medicine and CS), where structured multi-entity relationships are crucial. These findings suggest that the relational inductive biases encoded by the hypergraph representation transfer effectively across specialized verticals not seen during training.
>
> - Collectively, these experiments demonstrate that Graph-R1 offers superior cross-domain robustness, **strong transferability to unseen fields**, and stable zero-shot performance, significantly outperforming chunk-based RAG baselines across both **horizontal (general)** and **vertical (specialized)** domains.
>
> ---

---

> ### Author Response · Authors · 2025-11-21
> **Response to Reviewer ZyZx (Part 3/3)**
>
> ---
> **Q3: "Some claims are too strong. For example, the paper claims that existing GraphRAG all aim to gather sufficient knowledge in a single fixed retrieval. However, there are already some GraphRAG work such as HybGRAG [1] that leverages multi-retrieval strategy. The paper need to do a more comprehensive literature review.
> [1] HybGRAG: Hybrid Retrieval-Augmented Generation on Textual and Relational Knowledge Bases(https://aclanthology.org/2025.acl-long.43/) (Lee et al., ACL 2025)"**
>
> - Thank you for the insightful comment. We agree that our original wording **(“...they all aim to gather sufficient knowledge in a single fixed retrieval...”)** was too strong. We have revised the statement by replacing **“they all”** with **“most of them”** to avoid overgeneralization.
> - In addition, we have **substantially expanded the literature review** in the **Related Work section** to include recent GraphRAG variants that support multi-step or hybrid retrieval, including HybGRAG[1], Youtu-GraphRAG[2], LinearRAG[3], GraphRAG-R1[4], E²GraphRAG[5], GraphSearch[6], GFM-RAG[7], and G-reasoner[8].
>
> >**References:**
> [1] Lee, et al. "Hybgrag: Hybrid retrieval-augmented generation on textual and relational knowledge bases." Proceedings of the 63rd Annual Meeting of the Association for Computational Linguistics (Volume 1: Long Papers). 2025.
>  [2] Dong, et al. "Youtu-graphrag: Vertically unified agents for graph retrieval-augmented complex reasoning." arXiv preprint arXiv:2508.19855 (2025).
> [3] Zhuang, et al. "LinearRAG: Linear Graph Retrieval Augmented Generation on Large-scale Corpora." arXiv preprint arXiv:2510.10114 (2025).
> [4] Yu, et al. "Graphrag-r1: Graph retrieval-augmented generation with process-constrained reinforcement learning." arXiv preprint arXiv:2507.23581 (2025).
> [5] Zhao, et al. "E^ 2GraphRAG: Streamlining Graph-based RAG for High Efficiency and Effectiveness." arXiv preprint arXiv:2505.24226 (2025).
> [6] Yang, et al. "GraphSearch: An Agentic Deep Searching Workflow for Graph Retrieval-Augmented Generation." arXiv preprint arXiv:2509.22009 (2025).
> [7] Luo, et al. "GFM-RAG: graph foundation model for retrieval augmented generation." arXiv preprint arXiv:2502.01113 (2025).
> [8] Luo, et al. "G-reasoner: Foundation Models for Unified Reasoning over Graph-structured Knowledge." arXiv preprint arXiv:2509.24276 (2025).
>
> - In the revised manuscript, the Related Work section now explicitly clarifies that most existing GraphRAG systems, such as GraphRAG, LightRAG, HippoRAG, and path-based variants—retrieve graph evidence in a single fixed round. We then add a dedicated sentence summarizing recent **multi-step GraphRAG approaches**, including **HybGRAG, Youtu-GraphRAG, GraphSearch**, explaining how they incorporate LLM-driven query refinement. This restructuring ensures that the reader clearly sees where our method fits: Graph-R1 **differs from these works** by providing **the first end-to-end learnable multi-turn retrieval policy** that interacts with graph structures, **rather than relying on prompt-driven refinement**.
> - This ensures a more comprehensive and balanced discussion of prior work, especially those adopting multi-retrieval strategies. We thank the reviewer for pointing this out, which helped us improve the clarity and fairness of our claims.
>
> ---
> At last, we sincerely appreciate your valuable feedback. We have carefully considered all your suggestions and substantially improved and updated the paper accordingly. If our responses and revisions address your concerns, we would be deeply grateful if you could kindly reconsider raising the score to 6 or above. Thank you very much!

---

> > ### Author Response · Authors · 2025-11-27
> > **Official Comment by Authors**
> >
> > Dear Reviewer ZyZx,
> >
> > Thank you again for the time and effort you’ve dedicated to reviewing our work. We have carefully addressed all raised concerns during the discussion phase and have also uploaded an updated version of the paper reflecting these clarifications.
> >
> > As the discussion period is nearing its close, we would greatly appreciate it if you could take a brief moment to review our responses and confirm whether they satisfactorily resolve your questions. If our clarifications have improved your confidence in the paper, we would be sincerely grateful if you could consider updating your score accordingly.
> >
> > Thank you once again for your thoughtful feedback and support.
> >
> > Warm regards,
> >
> > Authors of Graph-R1

---

### Official Review · Reviewer_ru5v · 2025-11-01

**Soundness:** 3
**Presentation:** 3
**Contribution:** 3
**Rating:** 6
**Confidence:** 4

**Summary:**

This paper proposes Graph-R1, a GraphRAG method that leverage LLMs as agent to interact with constructed knowledge hypergraph. The proposed method is trained via end-to-end reinforcement learning. Experiments on multiple dtasets demonstrate the proposed Graph-R1 outperforms bases while also achieve strong generalization ability.

**Strengths:**

1. The proposed Graph-R1 leverage LLMs as agent to interact with graph, which solve the issues of finxed retrieval process with only one-time interaction.

2. The experiments are conducted on multiple datasets and include recent baselines.

3. The authors also conduct detaield ablation studies and generalization comparision.

**Weaknesses:**

1. There is another work GraphRAG-R1 [1], which also use RL to solve the GraphRAG issues. The authors may compare the difference between these two methods.

2. What is the motivation to use Knowledge HyperGraph intead of KG?


[1] Yu, Chuanyue, et al. "GraphRAG-R1: Graph Retrieval-Augmented Generation with Process-Constrained Reinforcement Learning." arXiv preprint arXiv:2507.23581 (2025).

**Questions:**

1. Can the propsoed method be applied to KGQA, which contains the explicit KG.
2. Is the <think> steps necessary for RL?
3. Why does GRPO significantly outperform PPO?

---

> ### Public Comment · ~Saptarshi_Sengupta1 · 2025-11-13
> **Not an author - just someone who's interested in the work**
>
> Dear Reviewer,
>
> I am interested in this work & hence commenting to aid the discussion/provide my PoV & not trying to interfere with the author's thought process.
>
> 1. The other paper you mentioned was written at roughly the same time as their work. As such, they are parallel & there was no way for this team to know about the other.
>
> 2. I agree with your point about the knowledge hypergraph. I think they claim that it's more "lightweight" & hence opt to use it instead of a regular KG. I agree with you that it's not super clear. Perhaps it's just another way of organizing the information.

---

> > ### Author Response · Authors · 2025-11-21
> > **Response to Reader**
> >
> > Dear Reader,
> >
> > Thank you very much for your interest in our work and taking part in the discussion. Below, we respectfully provide our detailed responses to address your comments.
> >
> > 1. Thank you for helping clarify this point. Yes, GraphRAG-R1 was developed during the same period as our work. We appreciate you pointing this out in the discussion.
> >
> > 2. As we explained in our response to W1, our use of a Knowledge HyperGraph is not merely for lightweight organization but for more **complete n-ary knowledge representation**, which directly benefits agentic RL. **A more expressive knowledge representation raises the RL performance ceiling**. We provide qualitative justification in **Proposition 1** and **Appendix B.1**, and quantitative evidence in **Figure 5(b)**, where HyperGraph-based environments outperform standard binary KGs.

---

> ### Author Response · Authors · 2025-11-21
> **Response to Reviewer ru5v (Part 1/2)**
>
> Thank you very much for your time and effort in reviewing our paper. We sincerely appreciate your feedback. Below, we respectfully provide our detailed responses to address your concerns.
>
> ---
> **W1: "There is another work GraphRAG-R1 [1], which also use RL to solve the GraphRAG issues. The authors may compare the difference between these two methods."**
>
> - We thank the reviewer for pointing out GraphRAG-R1 [1]. Below we summarize the **similarities** and **differences** between the two approaches.
> - First, in terms of shared motivation, both GraphRAG-R1 and our Graph-R1 aim to **address the limitations of traditional GraphRAG**, especially the inability of single-shot retrieval to handle complex multi-hop reasoning. Both works introduce reinforcement learning (RL) to improve retrieval–reasoning synergy and enhance multi-step decision making.
> - However, the two approaches differ substantially in methodology and design philosophy. GraphRAG-R1 focuses on learning retrieval-tool invocation, and its RL training **requires a cold-start supervised fine-tuning stage** to teach the model the correct output format before RL can be applied. Its RL objective is then shaped by **process-constrained rewards** to balance shallow vs. excessive retrieval.
> - In contrast, Graph-R1 introduces an explicit agentic graph environment and models the entire reasoning process—thinking, query formulation, multi-turn retrieval, and answering—as a unified end-to-end RL problem. Importantly, Graph-R1 **does not require cold-start SFT**: the agent learns directly from outcome-directed RL over the knowledge hypergraph, leveraging the structured environment to stabilize training without format-priming or multi-stage reward schedules. Furthermore, Graph-R1 performs RL inside a **lightweight knowledge hypergraph**, enabling multi-turn graph exploration, whereas GraphRAG-R1 does not employ hypergraph reasoning and instead augments conventional graph/textual retrieval.
> - We would also like to clarify that GraphRAG-R1 is a **concurrent work** developed approximately at the same time as ours, and therefore was not available during our initial submission.
> - Following the reviewer’s suggestion, we have added GraphRAG-R1 to the **Related Work section** and discussed its differences from Graph-R1 more explicitly.
>
> > **Reference:**
> [1] Yu, Chuanyue, et al. "GraphRAG-R1: Graph Retrieval-Augmented Generation with Process-Constrained Reinforcement Learning." arXiv preprint arXiv:2507.23581 (2025).
>
> ---
> **W2: "What is the motivation to use Knowledge HyperGraph intead of KG?"**
>
> - Thank you for the question. In our **Preliminaries**, we formalize GraphRAG as consisting of three essential components: graph construction, graph retrieval, and answer generation. This formulation makes it clear that an effective GraphRAG system needs both (i) **a stronger knowledge representation** during graph construction, and (ii) a representation that can truly support retrieval and generation in a way that **improves downstream reasoning**.
> - From this perspective, we choose Knowledge HyperGraphs instead of standard binary KGs for two reasons.
> - **First**, HyperGraphs offer **more complete knowledge representation**: many real-world facts are inherently n-ary, and forcing them into binary triples leads to semantic loss and fragmented relational structure. Hyperedges allow us to **keep these multi-entity relations intact**, preserving the coherence and completeness of each fact. This directly benefits both retrieval and reasoning.
> - **Second**, we are the first to treat a Knowledge HyperGraph as the agent environment in an end-to-end large-model reinforcement learning framework. This design allows the agent to interact with high-order relational structures during multi-turn reasoning, fully utilizing the knowledge graph instead of leaving it as a static lookup structure. Therefore, to **push both knowledge representation and RL-based reasoning to their full potential**, we adopt HyperGraphs.
> - Finally, as shown in **Figure 5(b)**, when using a standard binary KG versus a HyperGraph as the environment, the training curves clearly demonstrate that the **completeness of the knowledge representation improves the performance ceiling of RL**. This observation is also **theoretically** supported in **Proposition 1** and **Appendix B.1**.
>
> ---

---

> ### Author Response · Authors · 2025-11-21
> **Response to Reviewer ru5v (Part 2/2)**
>
> ---
> **Q1: "Can the propsoed method be applied to KGQA, which contains the explicit KG."**
> - Absolutely **yes**, the proposed method can be applied to KGQA.
> - KGQA and GraphRAG differ mainly in the **type of knowledge source** they operate on. KGQA systems are typically built on large, **explicit knowledge bases** such as **Freebase** or **Wikidata**, and existing approaches generally fall into two families: (1) retrieval-based KGQA, and (2) semantic-parsing-based KGQA.
> - For **retrieval-based KGQA**, such as **ToG [2]** and **RoG [3]**, the connection to our Graph-R1 framework is very direct. The KG is already a structured graph, so it can **naturally serve as the environment** for our RL-based agent. The agent can be trained end-to-end to perform multi-turn reasoning over this KG and generate the final answer. This is structurally very similar to how GraphRAG-enhanced generation works in our paper, except that the graph is now an explicit KB rather than a constructed one.
> - For **semantic-parsing–based KGQA**, questions are converted into **logical forms or executable queries** (e.g., SPARQL). Our Graph-R1 agentic framework is also compatible with this paradigm: each reasoning step currently generates a natural-language query, but this can be **replaced** with a logical-form query or executable KG query. This design is already aligned with prior KGQA work such as **ChatKBQA [4]**, where the model issues **structured queries** and receives the execution results. Moreover, under Graph-R1 framework, the agent can perform multi-turn refinement of such logical queries until the correct answer is obtained.
> - Therefore, **extending Graph-R1 to KGQA** is not only feasible but also a very **promising** direction: the agent can directly interact with an explicit KG, use structured queries, and benefit from multi-turn RL-driven reasoning to achieve more accurate KG question answering.
>
> > **References:**
> [2] Sun, et al. "Think-on-Graph: Deep and Responsible Reasoning of Large Language Model on Knowledge Graph." The Twelfth International Conference on Learning Representations.
> [3] LUO, et al. "Reasoning on Graphs: Faithful and Interpretable Large Language Model Reasoning." The Twelfth International Conference on Learning Representations.
> [4] Luo, et al. "Chatkbqa: A generate-then-retrieve framework for knowledge base question answering with fine-tuned large language models." Findings of the association for computational linguistics: ACL 2024. 2024.
>
> ---
> **Q2: "Is the [object Object] steps necessary for RL?"**
>
> - Sorry, it seems that the text shown as **“[object Object]”** in the review **did not display correctly**, so we are unable to determine what the reviewer was referring to. Could you please clarify which specific steps or components you are asking about? Once we have that information, we will be happy to provide a detailed response regarding their necessity for RL.
>
> ---
> **Q3: "Why does GRPO significantly outperform PPO?"**
>
> - The main reason is that PPO uses both a **policy model** and a **separate critic** (value-model) which must be optimized together. This dual-model setup **reduces training efficiency** and complicates training—especially in agentic RL setups with rich environment feedback and reasoning chains, where the critic struggles to evaluate “thinking” states and intermediate transitions.
> - In contrast, Group Relative Policy Optimization (GRPO) **omits the critic**: it samples a group of outputs per query, uses the group mean as the baseline, and directly optimizes the policy via relative advantages within that group. This critic-free design simplifies training and reduces error from value estimation, making it particularly **effective** for our agentic-environment reasoning scenario.
> - For more details, we find some knowledge source for reference:
>   - Shao et al., *DeepSeekMath: Pushing the Limits of Mathematical Reasoning in Open Language Models* (2024). (https://arxiv.org/abs/2402.03300)
>   - Blog “GRPO vs. PPO: A Simple, Clear Guide”. (https://aipapersacademy.com/deepseekmath-grpo/)
>
> ---
> At last, we sincerely appreciate your valuable feedback. We have carefully considered all your suggestions and substantially improved and updated the paper accordingly. If our responses and revisions address your concerns, we would be deeply grateful if you could kindly consider raising the score further to support our work. Thank you very much!

---

> > ### Comment · Reviewer_ru5v · 2025-11-25
> >
> > Thanks for the authors’ response. The “[object Object]” was caused by a copy-and-paste issue , and it should have been <think>...</think>. Sorry for the confusion.

---

> > > ### Author Response · Authors · 2025-11-25
> > >
> > > Thank you for the clarification. It seems the content still cannot be displayed correctly — it appears as “[object Object]...[object Object]” in our view.

---

> > ### Author Response · Authors · 2025-11-27
> > **Official Comment by Authors**
> >
> > Dear Reviewer ru5v,
> >
> > Thank you again for the time and effort you’ve dedicated to reviewing our work. We have carefully addressed all raised concerns during the discussion phase and have also uploaded an updated version of the paper reflecting these clarifications.
> >
> > As the discussion period is nearing its close, we would greatly appreciate it if you could take a brief moment to review our responses and confirm whether they satisfactorily resolve your questions. If our clarifications have improved your confidence in the paper, we would be sincerely grateful if you could consider updating your score accordingly.
> >
> > Thank you once again for your thoughtful feedback and support.
> >
> > Warm regards,
> >
> > Authors of Graph-R1

---

### Official Review · Reviewer_SR8i · 2025-11-01

**Soundness:** 3
**Presentation:** 3
**Contribution:** 2
**Rating:** 4
**Confidence:** 4

**Summary:**

This paper introduces Graph-R1, an RL-post-trained agentic GraphRAG framework. It operates on a constructed knowledge hypergraph and answers queries in a multi-turn fashion, i.e., it repeatedly performs a "think-retrieve-rethink-generate" loop within the knowledge hypergraph environment. Graph-R1 is trained using GRPO, with rewards based on format correctness and answer accuracy. Comprehensive empirical studies are conducted over multiple QA datasets, demonstrating the effectiveness of Graph-R1.

**Strengths:**

1. RL post-training with a multi-turn query and retrieval design makes sense.
2. The empirical studies are comprehensive.
3. The paper is well written and nicely illustrated.

**Weaknesses:**

1. The technical contributions seem limited: multi-turn action loop + GRPO + downstream task accuracy reward + format correctness reward are not particularly new. The authors could further clarify the core contributions.
2. The propositions appear decorative rather than informative.
3. I am curious about how the knowledge hypergraphs are built. It is not entirely clear to me from Sec. 2.1. Could the authors further clarify how this step is performed?
4. I am also interested in the generalization behavior of the trained policy. What would happen if the model were trained on all datasets: could it then achieve a universally effective (and maybe better) policy across all test sets?

**Questions:**

See above.

---

> ### Author Response · Authors · 2025-11-21
> **Response to Reviewer SR8i (Part 1/3)**
>
> Thank you very much for your time and effort in reviewing our paper. We sincerely appreciate your feedback. Below, we respectfully provide our detailed responses to address your concerns.
>
> ---
> **Q1: "The technical contributions seem limited: multi-turn action loop + GRPO + downstream task accuracy reward + format correctness reward are not particularly new. The authors could further clarify the core contributions."**
>
> Thank you for the thoughtful feedback. To clarify our contributions, we summarize them from the following **five perspectives**:
>
> **a. Motivation Level:**
>   - Prior GraphRAG systems rely on passive, one-shot retrieval, and **LLMs struggle to interpret retrieved graph structures**, often performing worse than chunk-based RAG.
>   - We identify GraphRAG’s **structural bottlenecks** (**high-cost** graph construction, **fixed** retrieval without reasoning, and a **gap** between graph representations and LLM reasoning), and want to **construct the first learnable, multi-turn agent that actively explores graph structures**, enabling reasoning over structured knowledge hypergraphs.
>
> **b. Technical Level:**
> - GraphRAG comprises **three core modules**—**construction**, **retrieval**, and **generation**. Addressing **key challenges in each part**, we introduce innovations that build a **learnable “data–knowledge dual-driven” GraphRAG system**.
> - For **Construction (Knowledge Construction and Agent Initialization)**, we propose a **lightweight n-ary hypergraph construction pipeline** and, for the first time, **formulate the hypergraph as an interactive agent environment** for structural interaction.
> - For **Retrieval (Knowledge Reasoning via Multi-Turn Graph Interaction)**, we develop a **dual-path retrieval mechanism fused via Reciprocal Rank Fusion (RRF)**, combining entity-based and semantic hyperedge-based signals to produce **stable, high-quality subgraph evidence**.
> - For **Generation (Outcome-Directed End-to-End Reinforcement Learning)**, we design a **specialized, outcome-aligned reward scheme** that couples **format-level structural validity** with **answer-level correctness**, ensuring reliable multi-step reasoning.
> - Together, these innovations yield the **first unified, interactive, and optimizable GraphRAG system**, with the effectiveness of each module validated in **Sections 5.4, 5.5, and 5.6**.
>
> **c. Theoretical Level:**
> - We show that RL not only improves answer quality but also refines retrieval trajectories, significantly **narrowing the gap** between **graph structures** and **LLM generation**.
> - **Proof 1: Graph-structured knowledge boosts agent accuracy by richer representation.**
> We support this by modeling the hypergraph as a more informative state space and observing consistent accuracy gains over text-based baselines.
> - **Proof 2: Multi-turn interaction with the graph environment improves retrieval efficiency.**
> We justify this through a state-update view of iterative refinement and empirically by reduced retrieval volume with higher answer quality.
> - **Proof 3: End-to-end RL bridges the gap between graph-based knowledge and language.**
> We validate this through a coupled optimization formulation and improved performance over non-learnable or decoupled pipelines.
> - To the best of our knowledge, this is the first work to **theoretically analyze** and **empirically validate** RL-driven reasoning over knowledge hypergraphs, which has not been explored in previous GraphRAG, Search-R1, or RL-RAG research.
>
> **d. Code Level:**
> - Graph-R1 is **the first open-source end-to-end trainable GraphRAG+RL framework**, supporting graph construction, retrieval, multi-turn agent rollouts, and GRPO training.
> - Unlike prompt-based GraphRAG systems, our framework enables **easily training an agent** that **actively operates within a graph**, providing substantial community value beyond a single algorithm.
>
> **e. Promising Level:**
> - Where current GraphRAG **relies on prompt engineering and long-context inference**, Graph-R1 shows that models can learn to use structured knowledge via RL.
> - This **neuro-symbolic direction** transforms graph knowledge from a passive context source into an active reasoning enhancer, opening a **scalable and learnable** pathway for future GraphRAG systems.
>
> In summary, Graph-R1 introduces **the first end-to-end learnable agentic GraphRAG framework** that actively interacts with hypergraph-structured knowledge, enabling the model to autonomously explore, select, and reason over structured evidence. By unifying hypergraph construction, multi-turn retrieval, and RL-driven reasoning within a single trainable system, our work demonstrates a scalable pathway for models to learn to use structured knowledge rather than rely on handcrafted prompts or fixed retrieval. We believe this establishes a solid foundation for future research on **learnable, neuro-symbolic GraphRAG systems**, and we hope this clarification helps highlight the significance of our contributions.
>
> ---

---

> ### Author Response · Authors · 2025-11-21
> **Response to Reviewer SR8i (Part 2/3)**
>
> ---
> **Q2: "The propositions appear decorative rather than informative."**
> - Thank you for pointing this out. We would like to clarify that **the propositions are not decorative**; they serve a specific **theoretical purpose** in our framework.
> - Our method consists of **three major components**, each explicitly designed to resolve one of the **three key challenges** of GraphRAG (high-cost construction, fixed retrieval, and the graph–language gap). For each component, we introduce a proposition that formalizes the expected effect of that module on the overall system behavior. These propositions are not merely descriptive statements, but are supported in two complementary ways:
> 	1.	**Quantitatively** — Each proposition corresponds to an experimental result validating the claimed effect (e.g., improvements in representation quality, retrieval efficiency, or graph–language alignment).
> 	2.	**Qualitatively** — We provide a formal analysis in the appendix showing why and under what conditions the component contributes to improved reasoning or retrieval performance.
> - Thus, the propositions serve as **bridges connecting the motivation, methodology, and empirical findings**. They are included to make the paper’s logical structure explicit and to ensure that each module’s contribution to addressing GraphRAG’s challenges is grounded in **both theory and experiment**.
>
> ---
> **Q3: "I am curious about how the knowledge hypergraphs are built. It is not entirely clear to me from Sec. 2.1. Could the authors further clarify how this step is performed?"**
> - Thank you for the question. We are glad to clarify the hypergraph construction process. To address this, we **added a dedicated section “Details of Knowledge Hypergraph Construction”** in **Appendix C**, together with **Figure 11**, which visually illustrates the entire pipeline.
> - As shown in **Figure 11**, Graph-R1 constructs the Knowledge HyperGraph through a streamlined yet expressive three-stage pipeline:
>     - **First**, We segment raw documents into fixed-size textual units to maintain stable local semantics. Each document is divided using a 1200-token window with a 50-token overlap, ensuring adjacent chunks retain sufficient contextual continuity without losing cross-sentence information.
>     - **Second**, each chunk is processed using a GPT-4o-mini–based n-ary relation extraction module. We adopt the carefully designed prompt in **Appendix A.1**, **Figure 9**, guiding the model to identify entity sets and the higher-order relations among them. Each extracted relational fact becomes a hyperedge with an associated coherence score, transforming linear text into a structured hypergraph where nodes represent entities and hyperedges encode rich n-ary facts.
>     - **Third**, for semantic retrieval and reasoning, we compute dense embeddings for both entities and hyperedges using bge-large-en-v1.5. These form the entity vector base and hyperedge vector base, which serve as the foundation of Graph-R1’s retrieval module.
> - Compared with the original HyperGraphRAG implementation, we include several architectural and engineering optimizations (parallelization, prompt efficiency, storage layout, and vectorization). These reduce token usage and computation cost, resulting in a highly efficient construction pipeline, as summarized in **Table 3** of **Section 5.4**.
> - We hope this clarifies how the hypergraph is constructed and why the process is both efficient and expressive.
> ---

---

> ### Author Response · Authors · 2025-11-21
> **Response to Reviewer SR8i (Part 3/3)**
>
> ---
> **Q4: "I am also interested in the generalization behavior of the trained policy. What would happen if the model were trained on all datasets: could it then achieve a universally effective (and maybe better) policy across all test sets?"**
>
> - Thank you for the insightful question. We share the same curiosity about whether training on all datasets could lead to a universally effective policy. To rigorously investigate this, we explicitly **designed the Combined-Dataset Training Setting** (newly described in **Appendix D**), where Graph-R1 is trained on all six datasets jointly, while strictly controlling the training data volume to match the I.I.D. setting.
> - Below, we present **Table 5** in full to clearly show the results across the three training regimes (I.I.D., single O.O.D., and combined), covering six multi-hop and open-domain datasets:
>
> | Method         | 2Wiki EM/F1 | HotpotQA EM/F1 | Musique EM/F1 | NQ EM/F1 | PopQA EM/F1 | TriviaQA EM/F1 | Avg. EM/F1 |
> |----------------|-------------|----------------|----------------|----------|--------------|-----------------|-------------|
> | **Qwen2.5-3B-Instruct** (I.I.D. Training) |
> | Search-R1 (3B) | 31.25 / 38.04 | 38.28 / 43.84 | 3.91 / 7.65 | 24.22 / 37.96 | 33.59 / 38.67 | 40.62 / 47.99 | 28.65 / 35.69 |
> | Graph-R1 (3B) | **50.00 / 57.56** | **50.78 / 56.75** | **32.81 / 40.51** | **30.47 / 44.75** | 37.50 / **45.65** | **53.13 / 62.31** | **42.45 / 51.26** |
> | **Qwen2.5-3B-Instruct** (Single O.O.D. Training) |
> | Search-R1 (3B) | — / 24.30 | — / 29.40 | — / 5.64 | — / 25.46 | — / 26.59 | — / 40.29 | — / 25.28 |
> | Graph-R1 (3B) | — / 42.65 | — / 50.20 | — / 32.06 | — / 38.20 | — / 42.63 | — / 58.85 | — / 44.10 |
> | **Qwen2.5-3B-Instruct** (Combined-Dataset Training) |
> | Search-R1 (3B) | 25.00 / 29.68 | 36.72 / 41.59 | 10.16 / 15.48 | 21.09 / 34.78 | 35.16 / 39.07 | 42.97 / 50.92 | 28.52 / 35.25 |
> | Graph-R1 (3B) | 42.97 / 51.07 | 46.88 / 54.97 | 32.03 / 38.29 | 27.34 / 41.00 | **38.28** / 45.32 | 50.00 / 58.49 | 39.58 / 48.19 |
>
> - As described in **Appendix D.1**, we observe two key findings from these results in this table (**Table 5**):
>   1. First, Graph-R1 achieves consistent and clear improvements over Search-R1 across all datasets and all training settings. This confirms the advantage of hypergraph-structured retrieval, which provides more coherent and relationally grounded evidence than chunk-based retrieval, leading to stronger reasoning and higher EM/F1 scores.
>   2. Second, comparing the three regimes, the results follow a stable pattern: I.I.D. training performs best, combined-dataset training ranks second, and single O.O.D. training performs worst. This indicates that dataset-aligned supervision remains the most effective, but exposure to balanced multi-dataset data is still substantially better than training on a completely mismatched dataset.
>
> - In summary, **while combined-dataset training does not surpass I.I.D. training, it still yields a robust policy that generalizes well across diverse datasets, outperforming single O.O.D. training**. We believe these findings provide valuable insights into the generalization behavior of RL-trained GraphRAG agents.
>
> ---
> At last, we sincerely appreciate your valuable feedback. We have carefully considered all your suggestions and substantially improved and updated the paper accordingly. If our responses and revisions address your concerns, we would be deeply grateful if you could kindly reconsider raising the score to 6 or above. Thank you very much!

---

> > ### Author Response · Authors · 2025-11-27
> > **Official Comment by Authors**
> >
> > Dear Reviewer SR8i,
> >
> > Thank you again for the time and effort you’ve dedicated to reviewing our work. We have carefully addressed all raised concerns during the discussion phase and have also uploaded an updated version of the paper reflecting these clarifications.
> >
> > As the discussion period is nearing its close, we would greatly appreciate it if you could take a brief moment to review our responses and confirm whether they satisfactorily resolve your questions. If our clarifications have improved your confidence in the paper, we would be sincerely grateful if you could consider updating your score accordingly.
> >
> > Thank you once again for your thoughtful feedback and support.
> >
> > Warm regards,
> >
> > Authors of Graph-R1

---

### Author Response · Authors · 2025-11-28
**Author Final Remarks by Authors**

Dear PCs, SACs, ACs, and Reviewers,

We thank all reviewers for their constructive feedback. Below we briefly summarize how we addressed each set of comments.

---

**SR8i**

*Concerns:*
Clarify core contributions and the role of the propositions; explain hypergraph construction; analyze generalization when training on all datasets jointly.

*Our updates and clarifications:*
- Clarified how each proposition aligns with a specific module and experiment in **Section 4** and **Appendix B**.
- Added a detailed three-stage construction pipeline with **Figure 11** in **Appendix C**.
- Introduced the Combined-Dataset setting in **Appendix D.1** and reported full results in **Table 5** (I.I.D. / single O.O.D. / combined).

*Summary:*
These changes make the method’s structure, construction process, and policy generalization behavior explicit and directly address the reviewer’s questions.

---

**ru5v**

*Concerns:*
Relationship to concurrent GraphRAG-R1; motivation for using a knowledge hypergraph instead of a KG; applicability to KGQA; and GRPO vs. PPO.

*Our updates and clarifications:*
- Discussed similarities and differences to GraphRAG-R1 in **Section 2** (Related Work).
- Clarified in **Appendix B.1** / **Figure 5(b)** why n-ary hyperedges give more complete relational states for RL than binary KGs.
- Outlined how Graph-R1 can be adapted to KGQA (graph as environment, structured queries as actions) in our response.
- Explained our choice of GRPO and its advantages for long reasoning chains; the “[object Object]” issue is purely a display artifact.

*Summary:*
The revised text now clearly positions Graph-R1 among related RL-GraphRAG methods, justifies the hypergraph design, and shows the framework’s compatibility with KGQA.

---

**ZyZx**

*Concerns:*
Perceived limited novelty; O.O.D. setting focused on Wikipedia; need for a broader GraphRAG literature review, especially multi-retrieval work.

*Our updates and clarifications:*
- Reorganized the contribution description in **Section 1** and **Section 4** to emphasize the unified agentic GraphRAG framework rather than individual components.
- Added five non-Wikipedia domains (Medicine, Agriculture, CS, Legal, Mixed) in **Appendix D.2**, with cross-domain results in **Table 6**.
- Expanded **Section 2** to include recent multi-step GraphRAG related work (e.g., HybGRAG, Youtu-GraphRAG, LinearRAG, GraphRAG-R1, E²GraphRAG, GraphSearch, GFM-RAG, G-reasoner) and softened the wording from “all” to “most” existing GraphRAG in **Section 1**.

*Summary:*
These updates strengthen cross-domain evaluation and give a more balanced and complete positioning of Graph-R1 within the evolving GraphRAG landscape.

---

**g7wu**

*Concerns:*
How Graph-R1 differs from combining Search-R1 with HyperGraphRAG; interpretation of metrics and Figure 5(b); handling queries without explicit entities; and training-data fairness vs. Search-R1.

*Our updates and clarifications:*
- Clarified in **Section 4** that hypergraph construction, dual-path retrieval, and outcome-based RL are integrated into a single learnable decision loop, instead of being used as separate modules.
- Clarified in **Section 5.2** and the caption of **Figure 5(b)** the roles of EM/F1, R-S, G-E, and that **Figure 5(b)** shows RL training iterations, not inference steps.
- Emphasized in **Section 4.2** that Graph-R1 performs dual-path retrieval over entities and natural-language hyperedges, supporting broader, less entity-centric queries.
- Stated in **Appendix F** that Graph-R1 and Search-R1 are retrained from scratch on the same resampled data (5,120 train / 128 test per dataset).

*Summary:*
These clarifications explain the integrated nature of the framework, remove potential misunderstandings about the plots and metrics, and confirm both the ability to handle non-entity queries and fairness in training.

---

Overll, we believe these updates and responses effectively address all concerns and clearly highlight that Graph-R1 provides the first unified, learnable neuro-symbolic GraphRAG framework that improves construction, retrieval, and generation from both fully **experimental** and **theoretical** perspectives. Moreover, with our open-sourced codebase, users can easily build their own knowledge graphs or hypergraphs easily and train tailored agents that fully leverage them, offering substantial practical value and serving as a promising contribution for future research in GraphRAG technology.

Best,

Authors of Graph-R1

---

### Meta-Review · Area_Chair_j5JD · 2026-01-05

**Summary:**

Almost all reviewers commented that the novelty of this paper is enough. It looks like that this work is a combination of several existing techniques.

**Reviewer Concerns:**

Although the authors provided clarifications on the novelty issue, I think the high-level idea of this paper is indeed quite similar to some existing works.

**Reviewer Scores:**

Some reviewers may raise their score, as they mentioned an existing work which was actually done within the same period as this submission. This submission should not be criticized because of this work.

---

### Decision · Program_Chairs · 2026-01-26

Reject